# Development of WRF/CUACE v1.0 model and its preliminary application in simulating air quality in China

Lei Zhang[1], Sunling Gong[1]*, Tianliang Zhao[2]*, Chunhong Zhou[1], Yuesi Wang[3], Jiawei Li[4], Dongsheng Ji[3], Jianjun He[1], Hongli Liu[1], Ke Gui[1], Xiaomei Guo[5,6], Jinhui Gao[7], Yunpeng Shan[8], Hong Wang[1], Yaqiang Wang[1], Huizheng Che[1], Xiaoye Zhang[1]

[1] State Key Laboratory of Severe Weather & Key Laboratory of Atmospheric Chemistry of CMA, Chinese Academy of Meteorological Sciences, Beijing, 100081, China
[2] Climate and Weather Disasters Collaborative Innovation Center, Nanjing University of Information Science &Technology, Nanjing, 210044 China
[3] State Key Laboratory of Atmospheric Boundary Layer Physics and Atmospheric Chemistry, Institute of Atmospheric Physics, Chinese Academy of Sciences, Beijing, 100029, China
[4] CAS Key Laboratory of Regional Climate-Environment for Temperate East Asia (RCE-TEA), Institute of Atmospheric Physics, Chinese Academy of Sciences, Beijing, 100029, China
[5] Heavy Rain and Drought-Flood Disasters in Plateau and Basin Key Laboratory of Sichuan Province, Chengdu, 610072, China
[6] Weather Modification Office of Sichuan Province, Chengdu, 610072, China
[7] Department of Ocean Science and Engineering, Southern University of Science and Technology, ShenZhen, 518055, China
[8] Environment and Climate Sciences Department, Brookhaven National Lab, Upton, NY, USA

*Correspondence to*: Sunling Gong (gongsl@cma.gov.cn) and Tianliang Zhao (tlzhao@nuist.edu.cn)

**Abstract.** The development of chemical transport models with advanced physics and chemical schemes could improve air-quality forecasts. In this study, the China Meteorological Administration Unified Atmospheric Chemistry Environment (CUACE) model, a comprehensive chemistry module incorporating gaseous chemistry and a size-segregated multicomponent aerosol algorithm, was coupled to the Weather Research and Forecasting (WRF)-Chem framework using an interface procedure to build the WRF/CUACE v1.0 model. The latest version of CUACE includes an updated aerosol dry deposition scheme and the introduction of heterogeneous chemical reactions on aerosol surfaces. We evaluated the WRF/CUACE v1.0 model by simulating $PM_{2.5}$, $O_3$, $NO_2$ and $SO_2$ concentrations for January, April, July, and October (representing winter, spring, summer, and autumn, respectively) in 2013, 2015, and 2017 and comparing them with ground-based observations. Secondary inorganic aerosol simulations for North China Plain (NCP), Yangtze River Delta (YRD), and Pearl River Delta (PRD) were also evaluated. The model well captured the variations of $PM_{2.5}$, $O_3$, and $NO_2$ concentrations in all seasons in eastern China. However, it is difficult to accurately reproduce the variations of air pollutants over Sichuan Basin (SCB), due to its deep basin terrain. The simulations of $SO_2$ were generally reasonable in the NCP and YRD with the bias at -15.5 % and 24.55 %, respectively, while poor in the PRD and SCB. The sulfate and nitrate simulations were substantially improved by introducing heterogenous chemical reactions into the CUACE model (e.g., change in bias from $-95.0\%$ to 4.1% for sulfate and from 124.1% to 96.0% for nitrate in the NCP). Additionally, The WRF/CUACE v1.0 model was revealed with better performance in simulating chemical species relative to the coupled

Fifth-Generation Penn State/NCAR Mesoscale Model (MM5)-CUACE model. The development of the WRF/CUACE v1.0 model represents an important step towards improving air-quality modelling and forecasts in China.

## 1 Introduction

The atmosphere is an extremely complex reaction system in which a large number of chemical and physical processes occur at every moment. Numerical modelling has become an effective means to study atmospheric environmental changes and their mechanisms due to its capability at large spatial-temporal scales and with high resolution. Against the continuing rapid increase in fine particle pollution in China, chemical transport models (CTMs) have been developed in recent years and new physical and chemical atmospheric mechanisms have been presented, for instance, heterogenous chemical reactions, the production of secondary organic and inorganic aerosols, and dry deposition schemes. However, some of the mechanisms have yet to be well parameterized into CTMs for air-quality forecasts in China. Numerical modelling in combination with field observations and laboratory analyses is constantly improving our understanding of atmospheric physical and chemical processes. There is an urgent need to develop and improve CTMs to provide more powerful tools for studying the atmospheric environment, in particular for the mitigation of fine particle pollution in China.

Meteorological conditions is accepted as one of the main factors affecting atmospheric chemical processes and the aerial transport of noxious materials, and, in turn, chemical species can impact meteorological conditions by radiation feedback and cloud formation (Grell and Baklanov, 2011). Historically, CTMs were developed separately from meteorological models owing to the complexity of the atmosphere and the economics of computer calculations. Thus, CTMs were generally driven by meteorological datasets from a pre-run of the meteorological model. Information about the rapid meteorological processes, such as changes in wind direction and speed or the planetary boundary layer, are barely recorded by the low-temporal-resolution meteorological outputs (typically once or twice per hour), which may impact the accuracy of the air-quality forecasts. Coupled systems that realize the synchronous integration and two-way interactions of meteorology and chemistry are an important development for the traditional CTM approach to air-quality forecasting and there have been many endeavors devoted to this (Jacobson et al., 1996; Lin et al., 2020; Lu et al., 2020; Zhang et al., 2010).

To tackle serious air pollution in China and East Asia, with a particular focus on haze pollution forecasting, the China Meteorological Administration (CMA) has been developing the Chinese

Unified Atmospheric Chemistry Environment (CUACE) model, a chemistry module that can be driven by meteorological models. The CUACE has been integrated into the Fifth-Generation Penn State/NCAR Mesoscale Model (MM5) and the mesoscale version of the Global/Regional Assimilation and Prediction System (GRAPES, a meteorological model developed by CMA) to build a fog-haze forecasting system (An et al., 2016; Wang et al., 2015a; Zhou et al., 2012). Both of these coupled systems have been running operationally at national and provincial meteorological administrations since 2014, and have been used for air-quality assurance for many major events in China. However, active development of the MM5 model ended with version 3.7.2 in 2005, and it has been largely superseded by the Weather Research and Forecasting (WRF) model (Skamarock, 2008). The WRF model has been shown to have a better performance relative to the MM5 model due to its better numerical dynamic core and greater number of physical parameterization schemes, and it is now used as a host model for coupling with different CTMs for scientific research and air-quality forecasting, such as the WRF-Chem and WRF-CMAQ models (Grell et al., 2005; Wong et al., 2012). The WRF model has also been used to provide pre-run meteorological fields to drive models such as CAMx and FLEXPART, as well as to provide boundary and initial fields for local-scale models. Therefore, it is important to develop the CUACE module by coupling it with state-of-the-art meteorological models.

The chemical reaction mechanisms in the CUACE module, as well as in current CTMs, are proposed under clean conditions. In the context of composite air pollution in China, particularly during severe haze episodes with a rapid increase in fine particles ($PM_{2.5}$), their applicability needs to be improved. Heterogenous chemical reactions, mechanisms missing in current models, were revealed as a crucial factor to explain the dramatic increase of $PM_{2.5}$ during hazy days (Zheng et al., 2015), such as the heterogenous uptake of dinitrogen pentoxide at night (Wang et al., 2017), and the heterogeneous oxidation of dissolved $SO_2$ by $NO_2$ (Gao et al., 2016; Seinfeld and Pandis, 2012). Another process focused on here is the dry deposition of particles, where the difference between model predictions and field measurements appears greatest for vegetated canopies and for the accumulation size range of airborne particles. Ongoing research is investigating the factors that give rise to this discrepancy and providing new approaches to predicting the deposition (Hicks et al., 2016). However, few studies have incorporated these mechanisms into 3D CTMs (Wu et al., 2018).

The objectives of this study were to develop the CUACE module from three aspects: (1) introduce heterogenous reactions and update the dry deposition scheme of particles; (2) couple the CUACE to the WRF model to build the WRF/CUACE v1.0 system; and (3) evaluate the model against observations of surface air pollutants.

## 2 Model description

### 2.1 WRF model

The Advanced Research WRF version 3 (WRF-ARW) is used to simulate meteorological processes and advection of atmospheric components in the WRF/CUACE v1.0 model. The WRF-ARW is a state-of-the-science mesoscale meteorological model, making simulations that are based on actual atmospheric conditions or idealized conditions feasible (Langkamp and Böhner, 2011). The equation set for the WRF-ARW is fully compressible, Eulerian non-hydrostatic with a run-time hydrostatic option. It is conservative for scalar variables. The prognostic variables consist of velocity components $u$ and $v$ in Cartesian coordinates, vertical velocity $w$, perturbation potential temperature, perturbation geopotential, and perturbation surface pressure of dry air, as well as several optional prognostic variables depending on the model physical options (Skamarock et al., 2008; Wong et al., 2012).

### 2.2 CUACE module

The CUACE is a unified chemistry module, which treats most of the physical and chemical processes, except advection and convection processes that done by its host model. The main processes treated in CUACE module include emissions, gas chemistry, dry and wet deposition, vertical mixing, aerosol-cloud interaction, and clear-air (i.e., aerosol produced by chemical transformation of their precursors together with particle nucleation, condensation and coagulation) (An et al., 2016; Zhou et al., 2012; Gong et al., 2003).

The CUACE is typically configured with the second generation of the Regional Acid Deposition Model (RADM2) as its gas chemistry module, which represents 63 species through 21 photochemical reactions and 136 gas phase reactions. As the gaseous chemistry (RADM2) in the CUACE module is not computationally economic and it is hard coded, which means that it is not conducive to adapting chemical reactions in the future, the CBM-Z photochemical mechanism (Zaveri and Peters, 1999) with a better computational efficiency is added with the KPP protocol (Damian et al., 2002) to replace the RADM2 mechanism. CBM-Z mechanism contains 55 species, 114 reactions and 20 photochemical reactions. It is based on the widely used Carbon Bond Mechanism (CBM-IV) and uses the lumped structure approach for condensing organic species and reactions. CBM-Z extends the CBM-IV to include revised inorganic chemistry, explicit treatment of the lesser reactive paraffins, methane and ethane, revised treatments of reactive paraffin, olefin, and aromatic reactions, inclusion of alkyl and acyl peroxy radical interactions and their reactions with

NO$_3$, inclusion of organic nitrates and hydroperoxides, and revised isoprene chemistry. Currently, stratospheric chemistry is not included in the CUACE module. Species (i.e, CH$_4$, CO, O$_3$, NO, NO$_2$, HNO$_3$, N$_2$O$_5$ and N$_2$O) above a specified pressure level are fixed to climatological values. Between the specified pressure level and the tropopause level, the species was relaxed with a 10-day relaxation factor.

The Canadian Aerosol Module (CAM) (Gong et al., 2003) is adopted as its aerosol module. There are totally seven types of aerosols treated in CAM, i.e. black carbon, primary organic carbon, sulfates, nitrates, ammonium, soil dust, and sea salts. The sea salt emissions are calculated online using the parametrization scheme developed by Gong et al. (2003). Soil dust emissions are simulated using the Marticorena–Bergametti–Alfaro scheme (Alfaro and Gomes, 2001; Marticorena and Bergametti, 1995). With the exception of ammonium, the aerosol size spectrum is divided into 12 bins with fixed boundaries of 0.005–0.01, 0.01–0.02, 0.02–0.04, 0.04–0.08, 0.08–0.16, 0.16–0.32, 0.32–0.64, 0.64–1.28, 1.28–2.56, 2.56–5.12, 5.12–10.24, and 10.24–20.48 μm. The detailed description of aerosol physical and chemical processes in the CAM module could be found in Gong et al. (2003).

## 3 Development of the CUACE module

### 3.1 Update with particle dry deposition scheme

The CUACE module currently parameterizes particle dry deposition velocity according to the method of Zhang et al. (2001) (Z01), which tends to overestimate the dry deposition, especially for fine particles (Petroff and Zhang, 2010). In this study, we use the scheme developed by Petroff and Zhang (2010) (PZ10) to replace the original scheme in the CUACE module. The most significant difference between the Z01 and PZ10 scheme is the treatment of $Rs$, which stands for the dry velocity contributed by surface resistance, consisting of Brownian diffusion, turbulent impaction, interception and rebound. According to the study of Wu et al., (2018), dry deposition velocity of fine particles is strongly affected by the Brownian diffusion and turbulent impaction. Thereby, it could be inferred that the Z01 scheme is prone to overestimate the effect of Brownian diffusion and turbulent impaction. In a recent study by Emerson et al. (2020), with observationally constrained approach, the Z01 scheme was revised to be with weaker effect of Brownian diffusion, and as a result, got better performance in simulating the dry deposition velocity of fine particles.

Both of the Z01 and PZ10 schemes use the "resistance" analogy, but with quite different formulas. The PZ10 scheme improved the surface resistance and collection efficiency of the Z01

scheme to overcome the problem of overestimating the dry deposition velocity of fine particles. The PZ10 scheme is detailed as follows:

$$V_d = V_{drift} + \frac{1}{R_a + R_s} \tag{1}$$

Here $V_d$ is the dry deposition velocity; $V_{drift}$ represents drift velocity, which is equal to the sum of gravitational settling and phoretic velocity and is expressed as

$$V_{drift} = V_g + V_{phor} \tag{2}$$

where $V_g$ is the gravitational settling velocity and $V_{phor}$ accounts for the phoretic effects that are related to differences in temperature, water vapor, or electricity between the collecting surfaces

and the air (Wu et al., 2018).

    The aerodynamic resistance $(R_a)$ and surface resistance $(R_s)$ are calculated differently for vegetated and unvegetated surfaces. For vegetated surfaces, $R_a$ is parameterized as

$$R_a = \frac{1}{\kappa * u_*}\left[\ln\left(\frac{z_R - d}{h - d}\right) - \Psi_h\left(\frac{z_R - d}{L_O}\right) + \Psi_h\left(\frac{h - d}{L_O}\right)\right] \tag{3}$$

where $\kappa$ is the von Karman constant (0.4), $u_*$ is the friction velocity above canopy, $z_R$ is the

reference height, $h$ is the canopy height, $d$ is the displacement height of the canopy, $L_O$ is the Obhukov length, and $\Psi_h$ is the integrated form of the stability function for heat.

    Surface resistance $(R_s)$ is generally expressed as the reciprocal of the surface deposition velocity $(V_{ds})$, which is parameterized as

$$V_{ds} = u_* E_g \frac{1 + \left[\frac{Q}{Q_g} - \frac{\alpha}{2}\right]\frac{tan(h\eta)}{\eta}}{1 + \left[\frac{Q}{Q_g} + \alpha\right]\frac{tan(h\eta)}{\eta}} \tag{4}$$

where $E_g = E_{gb} + E_{gt}$ is the total collection efficiency on the ground below the vegetation. $E_{gb}$ and $E_{gt}$ represent Brownian diffusion and turbulent impaction, respectively. $E_{gb}$ is parameterized as

$$E_{gb} = \frac{Sc^{-\frac{2}{3}}}{14.5}\left[\frac{1}{6}\ln\frac{(1+F)^2}{1-F+F^2} + \frac{1}{\sqrt{3}}Arctan\left(\frac{2F-1}{\sqrt{3}}\right) + \frac{\pi}{6\sqrt{3}}\right]^{-1} \tag{5}$$

    where $F$ is a function of the Schmidt number $(S_c)$ and is parameterized as $F = Sc^{\frac{1}{3}}/2.9$. $E_{gt}$ is

expressed as

$$E_{gt} = 2.5 \times 10^{-3} C_{IT} * \tau_{ph}^{+2}, \tag{6}$$

where $C_{IT}$ is a constant taken as 0.14 and $\tau_{ph}^+$ is a function of non-dimensional relaxation time of the particle (Petroff et al., 2010).

    In equation (4), the non-dimensional timescale parameter, $Q$, represents the ratio of turbulent

transport timescale to vegetation collection timescale, and $Q_g$ is the analogy of $Q$ used for the

transfer to the ground. $Q \ll 1$ characterizes a situation where turbulent mixing is efficient and the transfer of particles is limited by the collection efficiency on leaves. Meanwhile, $Q \gg 1$ corresponds to a situation where particles are efficiently collected by leaves and transfer of turbulent mixing is limited. $Q$ and $Q_g$ are defined as:

$$Q = \frac{LAI * E_T * h}{l_{mp}(h)} \tag{7}$$

$$Q_g = \frac{E_g * h}{l_{mp}(h)} \tag{8}$$

where $LAI$ is the two-sided leaf area index, $E_T$ is the total collection efficiency by various physical processes, and $l_{mp}$ is the mixing length for particles. $E_T$ is expressed as:

$$E_T = \frac{U_h}{u_*}(E_B + E_{IN} + E_{IM}) + E_{IT} \tag{9}$$

where $U_h$ is the horizontal mean wind speed at canopy height $h$; and $E_B$, $E_{IN}$, $E_{IM}$, and $E_{IT}$ are the collection efficiencies by Brownian diffusion, interception, inertial impaction, and turbulent impaction, respectively. The term $\eta$ is taken as

$$\eta = \sqrt{\frac{\alpha^2}{4} + Q} \tag{10}$$

where $\alpha$ is the aerodynamic extinction coefficient, and is expressed as

$$\alpha = \left(\frac{k_x * LAI}{12k^2\left(1-\frac{d}{h}\right)^2}\right)^{\frac{1}{3}} \phi_m^{\frac{2}{3}} \left(\frac{h-d}{L_O}\right) \tag{11}$$

where $k_x$ is the inclination coefficient of the canopy elements and $\phi_m$ is the non-dimensional stability function for momentum.

For non-vegetated surfaces, the aerodynamic resistance $Ra$ is calculated as

$$Ra = \frac{1}{\kappa u_*}\left[\ln\left(\frac{z_R-d}{z_0}\right) - \Psi_h\left(\frac{z_R-d}{L_O}\right) + \Psi_h\left(\frac{z_0}{L_O}\right)\right] \tag{12}$$

and the surface deposition velocity $V_{ds}$ is expressed as

$$V_{ds} = u_*(E_{gb} + E_{IT}) \tag{13}$$

## 3.2 Introduction of heterogeneous chemistry

The study of heterogeneous chemical reactions mostly focuses on the surface of dust aerosols, but the parameterization schemes of heterogeneous chemical reactions on different types of aerosol have not been well established (Zheng et al., 2015). The following are the heterogeneous chemical reactions on aerosol surfaces that added to the CUACE module in this study ("Aerosol" in the reactions stands for all the aerosols in the model):

$$\text{H}_2\text{O}_2 \text{ (gas)} \xrightarrow{\text{Aerosol}} \text{Products} \qquad\qquad (14)$$

$$\text{HNO}_3 \text{ (gas)} \xrightarrow{\text{Aerosol}} 0.5\text{NO}_3^- + 0.5\text{NO}_x \text{ (gas)} \qquad (15)$$

$$\text{HO}_2 \text{ (gas)} + \text{Fe(II)} \rightarrow \text{Fe(III)} + \text{H}_2\text{O}_2 \qquad\qquad (16)$$

$$\text{N}_2\text{O}_5 \text{ (gas)} \xrightarrow{\text{Aerosol}} 2\,\text{NO}_3^- \qquad\qquad (17)$$

$$\text{NO}_2 \text{ (gas)} \xrightarrow{\text{Aerosol}} \text{NO}_3^- \qquad\qquad (18)$$

$$\text{NO}_3 \text{ (gas)} \xrightarrow{\text{Aerosol}} \text{NO}_3^- \qquad\qquad (19)$$

$$\text{O}_3 \text{ (gas)} \xrightarrow{\text{Aerosol}} \text{Products} \qquad\qquad (20)$$

$$\text{OH (gas)} \xrightarrow{\text{Aerosol}} \text{Products} \qquad\qquad (21)$$

$$\text{SO}_2 \text{ (gas)} \xrightarrow{\text{Aerosol}} \text{SO}_4^{2-} \qquad\qquad (22)$$

Reactions (15) and (17)–(19) describe the formation of sulfate and nitrate on the surface of sand dust, and the other four reactions describe mineral aerosols as sinks of gaseous substances. In this study, these nine heterogeneous reactions were extended to all types of aerosol surface in the CUACE, referring to the approach of Zheng et al. (2015) for the CMAQ model. The first-order chemical kinetic equation for calculating the adsorption efficiency of a gas on an aerosol surface is:

$$\frac{dC_i}{dt} = -k_i C_i \qquad\qquad (23)$$

where $C_i$ represents the concentration of gas $i$ and $k_i$ is the pseudo-first-order rate constant and is supposed to be irreversible. The value of $k_i$ is defined referring to Jacob (2000) as:

$$k_i = \left(\frac{a}{D_i} + \frac{4}{v_i \gamma_i}\right)^{-1} A \qquad\qquad (24)$$

where $a$ is the aerosol diameter, $D_i$ is the diffusion coefficient for gas reactant $i$, $v_i$ is the mean molecule speed of gas reactant $i$, $\gamma_i$ is the uptake coefficient of the heterogeneous reaction for the gas reactant $i$, and $A$ is the surface area of aerosols in unit volume air. The value of $\gamma_i$ is obtained from previous laboratory studies (Table 1) and other parameters are calculated in the WRF/CUACE v1.0 model.

## 4 Coupling of the CUACE module with the WRF model

The coupling of the WRF/CUACE v1.0 model is based on the framework of WRF/Chem model and uses most of its existing infrastructure. WRF-Chem is a meteorology-chemistry coupled model. In the chemical module of the WRF-Chem, the processes are split to emissions, vertical mixing, dry deposition, convection, gas chemistry, cloud chemistry, aerosol chemistry and wet deposition, all of which are integrated in an interface procedure (chem_driver). Advection process is treated in the WRF model. Information, such as rainfall rates, vertical mixing coefficients and convective updraft properties, is provided by WRF to calculate the processes treated in the chemical module. WRF-Chem uses registry tools for automatic generation of application code. Physical and chemical

variables, as well as options of parameterization schemes are coded in files (such as registry.chem) in the directory of WRFV3/Registry, which provides the convenience for developers to add variables and options.

Following the registry tools in WRF-Chem model, a registry file (registry.cuace) is written to store the chemical variables and startup option of the CUACE module. The flow of the major process splitting in the coupled WRF/CUACE v1.0 model is illustrated in Fig. 1 with the structure of related subroutines given in Fig. S1 in the supplement. The WRF/CUACE v1.0 model uses several modules of the original WRF/Chem model, i.e., modules of advection, vertical mixing, convection, biomass emissions, anthropogenic gas emissions, photolysis and gas dry/wet deposition (Fig. S1). As described in Section 2.2, the CBM-Z mechanism is newly added with the KPP protocol (Damian et al., 2002) to replace the RADM2 mechanism in the original CUACE module. An interface procedure, cuace_driver, is designed to integrate the core sections of the aerosol physical and chemical processes of the CUACE module with the WRF framework (Fig. S1).

No spatial interpolation of the meteorological and chemical data is required as both the CUACE and the WRF models can be configured to the same gird configurations and coordinate systems. The feedback of chemical species on meteorology in the current WRF/CUACE version is not realized, but is under development and will be released in a future paper.

## 5 Performance of WRF/CUACE v1.0 in air-quality simulation

### 5.1 Model configuration

At present, there are four major polluted areas in China, namely, the North China Plain (NCP), the Yangtze River Delta (YRD), the Pearl River Delta (PRD), and Sichuan Basin (SCB). To include all these regions, the simulation area is configured as in Fig. 2. There are two domains in total. The boundary field of the inner domain is obtained by the interpolation of its outer domain. The outer region covers the whole of East Asia and its adjacent areas with a horizontal resolution of 54 km and a total of 120×110 grids centered at 30.46° N and 105.82° E. The inner region covers most of China on the east side of the Qinghai-Tibet Plateau with a horizontal resolution of 18 km and 193×175 grids. There are 32 vertical layers with the top pressure at about 100 hPa. The main physical and chemical options in the model are shown in Table 2. With WRF used in non-hydrostatic mode, we performed two simulations. One for January, April, July, and October in three years, 2013, 2015, and 2017, to evaluate the model on a long timescale, and one for three periods during which SIA observations were conducted (i.e., 5–16 January 2019 in Langfang, 3–29 December 2013 in Nanjing,

and 1–10 January 2017 in Chengdu), to investigate improvements in simulating SIA with heterogenous chemistry.

The model uses the FNL global reanalysis data of the NCEP (National Centers for Environmental Prediction) to provide the meteorological initial and boundary fields with spatial and temporal resolution of 6 h and $1°\times1°$, respectively. The initial and boundary chemistry conditions are based on the vertical profiles of $O_3$, $SO_2$, $NO_2$, VOCs (volatile organic compounds), and other air pollutants from the NOAA Aeronomy Lab Regional Oxidant Model (NALROM) (Liu et al., 1996).

Anthropogenic emissions are derived from the MIX emission inventory representative for 2010 (http://www.meicmodel.org/dataset-mix.html) (Li et al., 2017), which is an Asian anthropogenic emissions inventory developed for the third phase of the East Asian Model Comparison Plan (MICS-Asia III) and the United Nations Hemispheric Atmospheric Pollution Transport Plan (HTAP). The inventory provides monthly grid emission data with 0.25° spatial resolution for five emission sectors (electricity, industry, civil, transportation, and agriculture), including $PM_{2.5}$, $PM_{10}$, nitrogen oxides ($NO_x$), sulfur dioxide ($SO_2$), carbon monoxide (CO), $NH_3$, black carbon (BC), organic carbon (OC), and non-methane volatile organic compounds (NMVOCs). During the simulation span from 2013 to 2017, China carried out strict air pollution control measures, which had a considerable impact on anthropogenic emissions. To make the anthropogenic emissions more suitable for the real emissions scenarios in the simulated years, the emissions in mainland China were replaced with the MEIC emissions inventory representative for 2012, 2014, and 2016 to represent the emissions scenarios in 2013, 2015, and 2017, respectively. Figure S2 in the supplement shows the MEIC emissions of $PM_{2.5}$, $NO_x$, $SO_2$ and CO in the three years, from which it can be seen that anthropogenic emissions of $PM_{2.5}$, $SO_2$ and CO reduced remarkably from 2012 to 2016.

For the vertical interpolation, we used the settings of Wang et al. (2010) and Zhou et al. (2017). The industrial emissions were allocated as 50, 30, and 20% in layers one to three of the model, respectively, and the power plant emission sources were allocated as 14, 46, 35, and 5% in model layers two to five, respectively. The emissions from transportation, residential, and agriculture were 95% and 5%, respectively, in the first and second layers of the model. Then, the inventory was distributed into hourly emissions using the monthly, weekly, and hourly profiles established by Tsinghua University (2006). VOCs released from vegetation was calculated online using the MEGAN model (Guenther, 2006).

## 5.2 Evaluation against ground-based observations

### 5.2.1 Meteorological evaluation

The simulated hourly temperature at 2 m (T2), hourly relative humidity at 2 m (RH2) and hourly wind speed at 10 m (WS10) were selected for evaluation. Table S1 in the supplement shows the observation mean, simulation mean, correlation coefficient ($R$), MB, ME and RMSE of the meteorological fields in the NCP, YRD, PRD and SCB, respectively. The MB and RMSE for T2 vary from 0.48 to 1.14 °C and from 2.01 to 2.50 °C, respectively, indicating surface temperatures are slightly overestimated in the four regions. The $R$ value for T2, ranging from 0.88 to 0.93, indicates the variation trends are well captured by the model. The model underestimates RH2 in the four regions with the MB ranging from -6.22 to -14.30 % and the RMSE ranging from 13.95 to 18.77 %, which are comparable with previous studies in China (Wang et al., 2014; Gao et al., 2016). The RMSE for WS10 in the four regions vary from 1.47 to 1.61 m s$^{-1}$, fall within the "good" model performance criteria (little than 2 m s$^{-1}$) proposed by Emery et al. (2001). However, it should be noted that the $R$ for WS10 in the SCB is relatively poor, indicating the variation trends were not well captured. The simulations of T2 and RH2 in the SCB are relatively poor than other regions as well. For example, the $R$, MB and RMSE values of T2 in the SCB are 0.88, 1.52 °C and 2.50 °C, respectively, while the values in the other three regions vary from 0.91 to 0.93, 0.48 to 1.14 °C and 2.01 to 2.39 °C. Generally, the model performed best in the YRD, followed by the PRD and NCP, and performed worst in the SCB for meteorological fields.

## 5.2.2 Chemical evaluation

In view of the spatial-temporal differences in the haze pollution that occur in the four different regions (i.e. NCP, YRD, PRD, and SCB), here we assessed surface $PM_{2.5}$, $O_3$, $NO_2$ and $SO_2$ simulated in the WRF/CUACE v1.0 model by region and season. Figure 3 presents a comparison of the modelled and observed daily mean $PM_{2.5}$ concentrations in spring, summer, autumn, and winter in the four regions. Overall, the WRF/CUACE v1.0 model well captured the variations in the $PM_{2.5}$ concentration, but with different performance in different regions and seasons. The correlation coefficients ($R$) for the NCP, YRD, and PRD are mostly above 0.60 and passed the 99% significance test. The $R$ value between the YRD and PRD is the highest (generally higher than 0.65), followed by the NCP. The NCP, YRD, and SCB simulations in autumn and winter are generally better than that in spring and summer according to the $R$ values, while that in the PRD is the opposite with a better performance during spring and summer seasons. The simulations are relatively poor in the SCB, where the complex terrain poses great challenges to meteorological field simulations (Table S1 in the supplement).

It is noteworthy that the WRF/CUACE v1.0 model systematically underestimated the daily

PM$_{2.5}$ concentrations in the NCP when it exceeded about 200 μg m$^{-3}$, which mostly happened during winter (Fig. 4a). By comparing the time series of observations and simulations (not shown), we found that the underestimation mainly occurred in the period of heavy haze pollution in some cities (such as Shijiazhuang, Hengshui, Handan, etc.). Two factors might be responsible for this. One is the uncertainty of emission sources. The formulation of an accurate emissions source inventory is always

a difficult problem, especially in China. In the NCP, the seasonal difference in emission sources is substantial. A large number of unorganized loose coal combustion emissions during the winter heating season cannot be promptly accounted for by the emissions source inventory system, which increases the uncertainty of the local emission sources. The other factor might be problems in the chemical reaction mechanisms. The haze pollution study found that PM$_{2.5}$ was mainly composed of

secondary particulate matter, including sulfate, nitrate, ammonium salt, and SOA (Huang et al., 2014). During heavy haze episodes, the concentration of sulfate increased substantially, but its formation mechanism remains not well recognized. The main international atmospheric chemical models (such as CMAQ, WRF-Chem, CAMx, etc.) are also found to be not ideal enough to simulate sulfate and SOA during heavy haze pollution in North China. Zheng et al. (2015) and Gao et al.

(2016) initially added SO$_2$ heterogeneous processes in the CMAQ and WRF-Chem models, and the simulation results of sulfate improved. Although heterogeneous chemical reaction mechanisms are introduced in this study, the simulation effect of sulfate needs to be further evaluated, and the simulation of SOA is more challenging, involving thousands of VOC species and determination of their saturation, atmospheric oxidation, free radicals, acidity, and basicity. The development of a

volatility basis set (VBS) is a major breakthrough that treats the organic gas/particle partitioning with a spectrum of volatilities using a saturation vapor concentration as the surrogate of volatility (Ahmadov et al., 2012; Donahue et al., 2006; Wang et al., 2015b).

The WRF/CUACE v1.0 model was further evaluated using hourly PM$_{2.5}$ concentrations and $R$, mean bias (MB), mean error (ME), normalized mean bias (NMB), normalized mean error (NME),

mean fractional bias (MFB), and mean fractional error (MFE) (Table 3). As can be seen from Table 3, the correlation coefficients $R$ for the NCP, YRD, PRD, and SCB are 0.59, 0.71, 0.68, and 0.59, respectively, all of which passed the 99% significance test. The YRD has the best correlation, followed by the PRD. MB values reflect that the performance of the model is reasonable in all regions, among which those in NCP and PRD are the best, with the MB values reaching −5.0 and 5.3

380   μg m$^{-3}$, respectively. However, the MB values show that the simulated concentration of PM$_{2.5}$ in NCP during winter is generally underestimated by 45 μg m$^{-3}$ and overestimated by 33.9 μg m$^{-3}$. The dramatic positive bias in summer in the NCP is mainly due to the uncertainty in anthropogenic emissions. It is known that PM$_{2.5}$ concentration is mainly driven by primary emissions, meteorology

and chemical reactions. Table S2 in the supplement shows the statistical metrics for hourly
meteorological fields in winter and summer in the NCP. It can be seen that the bias of summer
meteorological fields is reasonable, and is comparable to those in winter (Table S2) as well as to
those in the YRD and PRD (Table S1), which indicate bias in meteorological fields is not the reason.
Additionally, In the YRD and PRD, where the uncertainties of anthropogenic emissions are generally
known as less than that of NCP, the bias of $PM_{2.5}$ between winter and summer are comparable (Table
3), implying chemical formation of $PM_{2.5}$ in summer is not overestimated by the WRF/CUACE v1.0
model.

From the point of view of relative deviation, the overall level of standard mean deviation NMB
in the NCP is slightly better than that in the YRD and PRD, but the seasonal difference is significant,
and the NMB values of the latter two (especially in the PRD) are more uniform in different seasons,
maintaining at about 20%, indicating that the simulation level of the model is relatively stable in the
region. The NMB of SCB is 12.2%, which is similar to that of NCP with a significant seasonal
difference (11.5% in winter and 60.4% in summer). The NMBs in the NCP, YRD and PRD are
basically the same, about 45%, slightly better than 50.3% in SCB.

Morris et al. (2005) provided a reference standard for MFB and MFE using hourly
concentrations of simulated and observed $PM_{2.5}$. The simulation performance is identified to be
excellent when MFB < 15% and MFE < 35%, identified to be good when MFB < 30% and MFE <
50%, and identified to be average when MFB < 60% and MFE < 75%, which are marked as bold,
normal, and italic font, respectively, in Table 3. It can be seen that simulations in the YRD and PRD
fall within the good level with the MFB/MFE reaching 21.1/42.9% and 8.6/40.1%, respectively. Both
reached excellent levels in winter, which are 8.5/34.1% and 5.5/34.4%. respectively, indicating that
the WRF/CUACE v1.0 model accurately captures the hourly variations of $PM_{2.5}$ in the two regions.
In the NCP region, the model still maintains a good simulation level (3.3/49.1%) in the area, with
obvious overestimates in summer but still maintaining an average level (44.9/56.3%). The SCB
region as a whole is at the average level (20.7/51.4%). The simulation of winter and spring is better
than that of spring and summer. The reason why the simulation in SCB is relatively poor is that its
topography is complex, which leads to inaccurate simulation of meteorological fields and further
affects the simulation of chemical species. In addition, the uncertainty of emission sources over there
is also a major factor (Zhang et al., 2019).

As a whole, the seven statistical error indicators $R$, MB, ME, NMB, NME, MFB, and MFE in
the four regions reached 0.63 (99% significance test), 2.7 μg m$^{-3}$, 33.3 μg m$^{-3}$, 2.8 %, 46.8 %,
10.6 %, and 46.2%, respectively, which showed that the WRF/CUACE v1.0 model can reasonably
reproduce the changes in $PM_{2.5}$.

Statistical metrics for $O_3$, $NO_2$ and $SO_2$, including index of agreement (IOA, see its definition in the supplement) (Willmott et al., 1980), NMB, and $R$, are shown in Table 4, along with a benchmark derived from the EPA (2005, 2007). In general, the $R$ values of $O_3$ and $NO_2$ in the four regions are about 0.6, which pass the 99% significance test. For $O_3$, NMBs indicate that the concentrations in the NCP, YRD, and PRD were well reproduced by simulations. The high consistency of the time series between the simulations and measurements was also reflected by the high values of IOA (>0.8). It should be noted that the NMB indicates that the $O_3$ concentrations in SCB were overestimated, which is also reflected in the scatter plot (Fig. 4d), partially due to the relatively poor simulation of meteorological fields (Table S1). As the precursor of $O_3$, simulation of $NO_2$ over the NCP, YRD, PRD, and SCB was acceptable, with the NMBs all falling within the benchmark and IOAs greater than 0.70. In general, the statistical metrics for $O_3$ and $NO_2$ are comparable with other studies (Gao et al., 2018; Hu et al., 2016). The variations of $SO_2$ in NCP and YRD were generally reproduced by the model with bias at -15.5 % and 24.55 %, respectively. However, in the PRD and SCB, $SO_2$ concentrations were substantially overestimated (Table 4 and Fig. 4k-l). As previous studies revealed, emissions of $SO_2$ in eastern China were overestimated by national emission inventories (Zhang et al., 2018; Zhou et al., 2019; Gao et al., 2016), which might partially contribute to the overestimation of $SO_2$ in YRD and PRD. On the basis of the above analysis results, the simulation results are satisfactory, with the exception of SCB.

### 5.3 Evaluation of SIA simulations with heterogeneous chemical reactions

Heterogeneous chemical reactions have been shown to have important effects on the formation of SIA, especially during severe haze events with high humidity (Li et al., 2011; Wang et al., 2006; Zhao et al., 2013). The ground observations of SIA from 5 to 16 January 2019 in Langfang (NCP), from 3 to 29 December 2013 in Nanjing (YRD), and from 1 to 10 January 2017 in Chengdu (SCB) were collected for the evaluation of SIA simulations. Following the model configurations in Section 4.2, we performed WRF/CUACE v1.0 simulations with (Exp_WH) and without (Exp_WoH) heterogenous chemistry on the three periods.

Figure 5 illustrates the hourly variations of observed SIA concentration from the Exp_WH and Exp_WoH experiments. For Langfang site, the simulation without heterogenous chemistry (Exp_WoH) barely capture the sulfate increase (Fig. 5a). This was substantially improved when heterogenous chemistry was included (Exp_WH), although some observed peak values are not well captured, such as those on 14 January. The overestimation of nitrate was also improved, with the NMBs changing from 124.1% to 96.0% (Fig. 5b). It should be noted that the responses of sulfate and

nitrate to heterogenous chemistry are inverse, which might be attributed to the complex thermodynamic processes of SIA formation (Zheng et al., 2015). Sulfate and nitrate will compete for ammonium, which is now the only cation currently in the CUACE model, resulting in less ammonium nitrate and more ammonium sulfate because of the more thermodynamically stable features of ammonium sulfate. As a result of the dramatical increase in sulfate in Exp_WH, the ammonium concentrations slightly increase relative to that in Exp_WoH to achieve anion–cation balance, which leads to more overestimations in the Exp_WH experiment (Fig. 5c). For Nanjing and Chengdu site, the underestimation of sulfate (Fig. 5d and 5g) and overestimation of nitrate (Fig. 5e and 5h) were also improved to varying degrees, with bias of sulfate changing from −95.3 % to -68.4 % in Nanjing and from -88.7 % to -80.1 % in Chengdu and the bias of nitrate changing from 83.0 % to 54.6 % in Nanjing and from 67.6 % to 23.5 % in Chengdu. Nonetheless, deviations in SIA simulations are still too large to neglect in those regions.

## 5.4 Comparison between the MM5/CUACE model and the WRF/CUACE v1.0 model

It is necessary to compare the MM5/CUACE model with the new WRF/CUACE model for the purpose of assessing the viability of the newly developed model. To this end, a simulation was performed using the MM5/CUACE model for a winter month, i.e., January 2013, during which a long-lasting haze event occurred in central and eastern China. The domain setting, anthropogenic emission inventory, initial and boundary fields of meteorology and chemistry are as the same as those of the WRF/CUACE in section 5.1. It should be known that the gas-phase chemistry mechanism and particle dry deposition scheme in MM5/CUACE model is RADM2 and Z01, respectively, that updated to CBM-Z and PZ10 in the new WRF/CUACE model. Physical parameterization used in the MM5/CUACE is shown in Table S3 in the supplement.

Figure 6 presents a comparison of the modelled and observed daily concentrations of $PM_{2.5}$, $O_3$, $NO_2$ and $SO_2$ in the four regions. It can be seen that the concentrations of $PM_{2.5}$, $NO_2$ and $SO_2$ simulated in WRF/CUACE are closer to the observations relative to those of MM5/CUACE model (change in bias from -23.0 % to -19.2 % for $PM_{2.5}$, from 14.7 % to -2.4 % for $NO_2$ and from -46.2 % to -37.5 % for $SO_2$). The daily variations of the three species are also relatively better captured by the WRF/CUACE model (reflected by the $R$ values changing from 0.45 to 0.62 for $PM_{2.5}$, from 0.41 to 0.49 for $NO_2$ and from 0.19 to 0.32 for $SO_2$). For $O_3$, the differences of statistical metrics between the two models are not obvious. The MM5/CUACE model performed with a slightly smaller bias of -10.7 % but with a lower R value of 0.50, which are 14.3 % and 0.55, respectively in the WRF/CUACE simulation. In summary, the new WRF/CUACE model performed better than the MM5/CUACE model

in simulating air pollutants.

## 6 Summary and future work

This study develops the chemical module CUACE by adding heterogenous chemical reactions
and introducing a particle dry deposition scheme developed by Petroff and Zhang (2010). The
CUACE module is then incorporated into the WRF-Chem model to build the WRF/CUACE v1.0
modelling system to take advantage of the better numerical dynamic core and the greater number of
physical parameterization schemes of the WRF model compared with the MM5 model.

We perform a three-year (2013, 2015, and 2017) model simulation using the WRF/CUACE v1.0
model to evaluate its performance on reproducing surface concentration variations of $PM_{2.5}$, $O_3$, and
$NO_2$, which are now the main pollutants in China. A heavy haze pollution event that occurred during
9–15 January 2019 in the NCP is also selected to evaluate the SIA simulations compared with
intensive ground SIA observations. The results show that WRF/CUACE v1.0 can well capture the
daily and hourly variations of $PM_{2.5}$, especially in the YRD and PRD regions throughout the three
years. For the NCP in winter, observed high concentrations larger than 200 μg m$^{-3}$ are not well
reproduced, which might be mainly due to uncertainties in the emissions inventory and the lack of
some chemical reactions in the model. For $NO_2$ and $O_3$, the model shows small biases in the NCP,
YRD, and PRD regions with correlation coefficients all larger than 0.60 and the NMBs all fall within
the EPA benchmark (2005, 2007). The model shows relatively notable biases in the SCB region
compared with the NCP, YRD, and PRD regions for the three pollutants, which may be mainly due to
the complex terrain in the SCB (Zhang et al., 2019) and insufficient meteorological data available for
the region for assimilation in the NCEP-FNL reanalysis data. Simulations of SIA are generally
improved, especially for sulfate in the NCP. However, large uncertainties remain in the mechanisms
of the heterogenous chemical reactions in the model, such as the determination of the uptake
coefficients, which is based on previous studies on dust surfaces.

There are still several limitations in the current version of the WRF/CUACE v1.0 model that
need to be addressed in future development. The feedback of particles, which can be divided into
direct and indirect effects, is recognized to be crucial in online coupled models, especially during
periods with high particle loading. Currently in the WRF-Chem model, the direct effects of aerosols
are processed following the methodology described by Ghan et al. (2001). Our future work will first
focus on implementing the direct effects of aerosols, i.e. radiation feedback, following the Mie
calculation to realize the direct aerosol forcing. The second step is to implement the VBS scheme to
add the missing processes of SOA, which has been implied to be a main cause in the underestimation

of OA formation (Gao et al., 2017; Heald et al., 2005; Spracklen et al., 2011). Although the original particle dry deposition scheme is updated with that developed by Petroff and Zhang (2010), it is difficult to evaluate whether the dry deposition process is improved as the limited technology of dry deposition observations restricts direct observations of particle dry deposition. With the observed $PM_{2.5}$ concentrations, model improvements with and without the updated dry deposition scheme are preliminary evaluated (Figure S3 in the supplement). With regards to particle dry deposition, our aim is to implement several schemes in the CUACE module, such as the schemes developed by Emerson et al. (2020), Zhang and He (2014), Zhang and Shao (2014), and Kouznetsov and Sofiev (2012), to evaluate uncertainties in the schemes on aerosol simulation, which might help the development of the particle dry deposition scheme.

### Code availability

The WRF/CUACE v1.0 model is open-source and can be accessed at a DOI repository https://doi.org/10.5281/zenodo.3872620. All source code and data can also be accessed by contacting the corresponding authors Sunling Gong (gongsl@cma.gov.cn) and Tianliang Zhao (tlzhao@nuist.edu.cn).

### Competing interests

The authors declare no competing interests.

### Author Contributions

Sunling Gong, Tianliang Zhao, Hong Wang, Huizheng Che and Xiaoye Zhang led the project. Lei Zhang, Sunling Gong, Chunhong Zhou and Hongli Liu developed the model code, with assistance from Jiawei Li, Jianjun He, Ke Gui and Yaqiang Wang. Lei Zhang performed the simulations and wrote the manuscript with suggestions from all authors. Yuesi Wang, Dongsheng Ji and Xiaomei Guo provided the data of secondary inorganic aerosols. Jinhui Gao and Yunpeng Shan contribute to data processing. All authors contributed to the discussion and improvement of the manuscript.

### Acknowledgements

This work is supported by the National Key Foundation Study Developing Programs (No. 2019YFC0214601), National Natural Science Foundation of China (No. 91744209, 41975131, 41705080), and the CAMS Basis Research Project (No. 2020Y001). We gratefully acknowledge the Atmosphere Sub-Center of Chinese Ecosystem Research Network (SCAS-CERN) for providing the data of secondary inorganic aerosols, and thank Prof. Leiming Zhang (Air Quality Research Division,

Science and Technology Branch, Environment Canada) for sharing the code of aerosol dry deposition scheme.

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

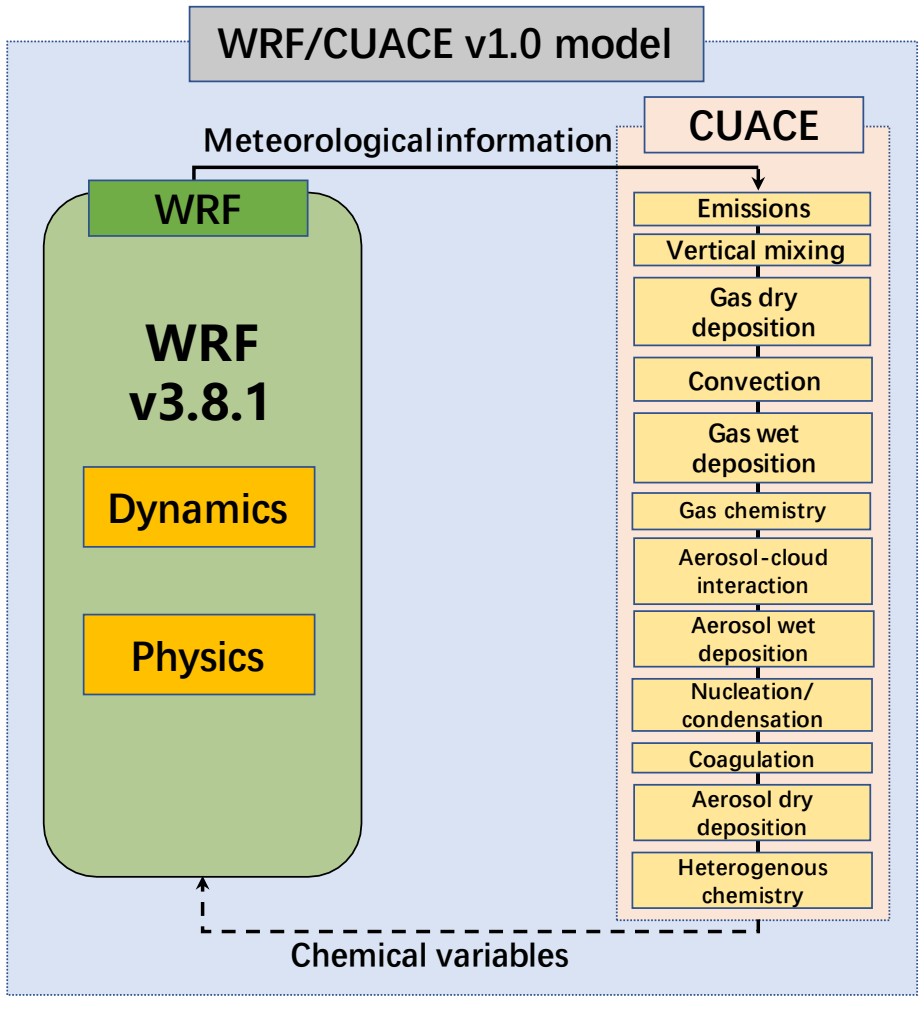

**Figure 1. Schematic of modules in the WRF/CUACE v1.0 system.**

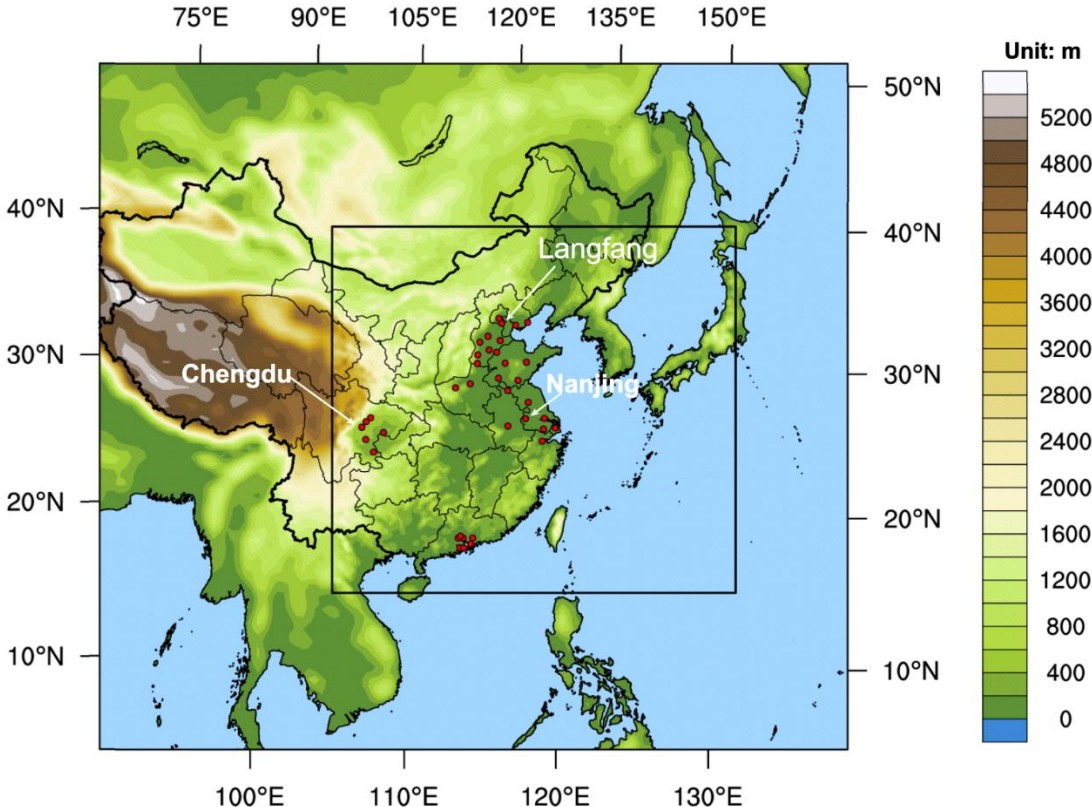

**Figure 2. Model domains with the terrain distribution. Red circles indicate the cities where the surface observations of air pollutants are used for model evaluation. The Langfang, Nanjing and Chengdu sites marked in this figure indicate where the SIA observations are collected for evaluation of SIA simulations.**

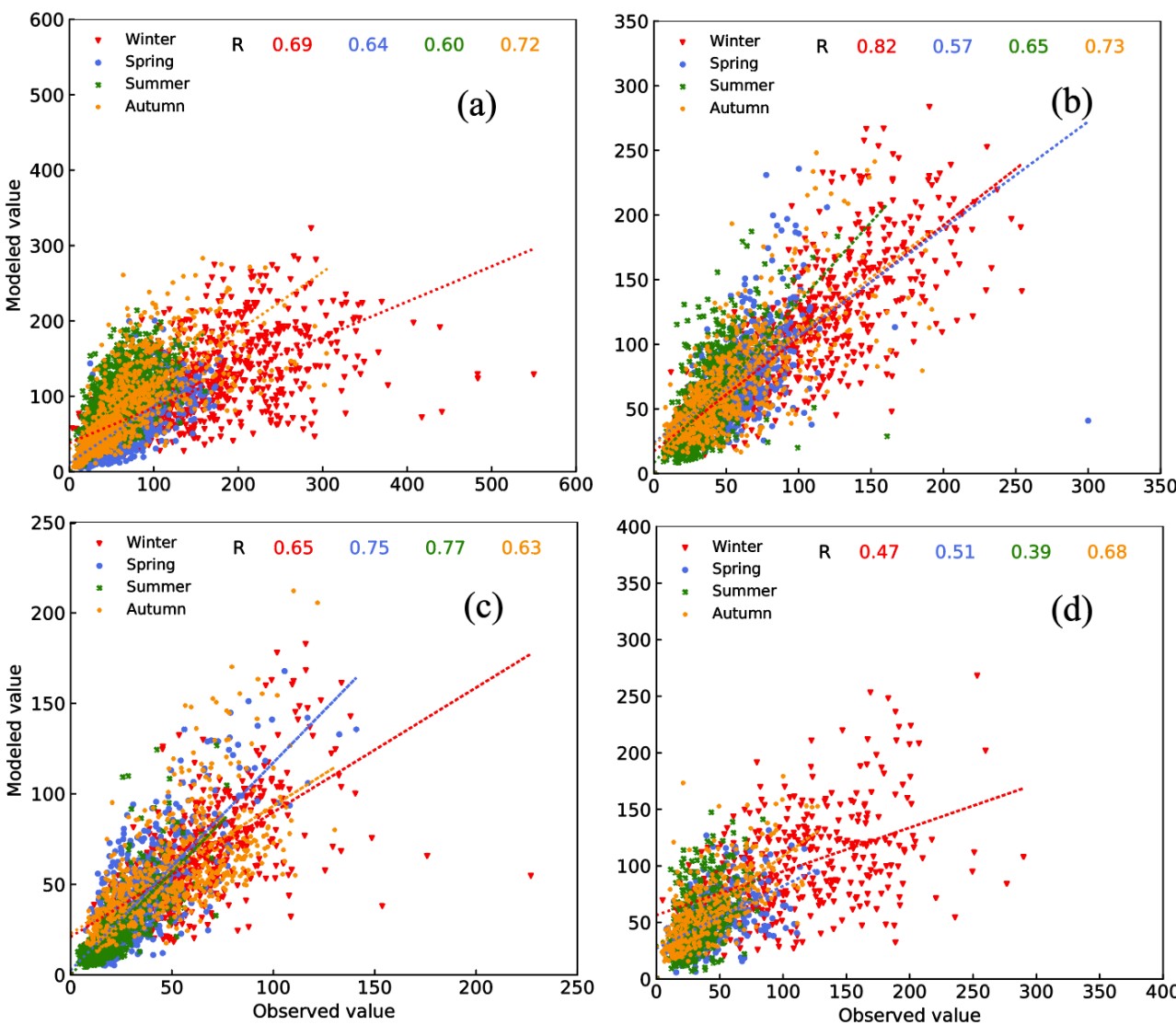

**Figure 3. Scatter plots and correlation coefficients of daily PM$_{2.5}$ concentrations (μg m$^{-3}$) between observed and simulated values in different seasons in the (a) NCP, (b) YRD, (c) PRD, and (d) SCB regions.**

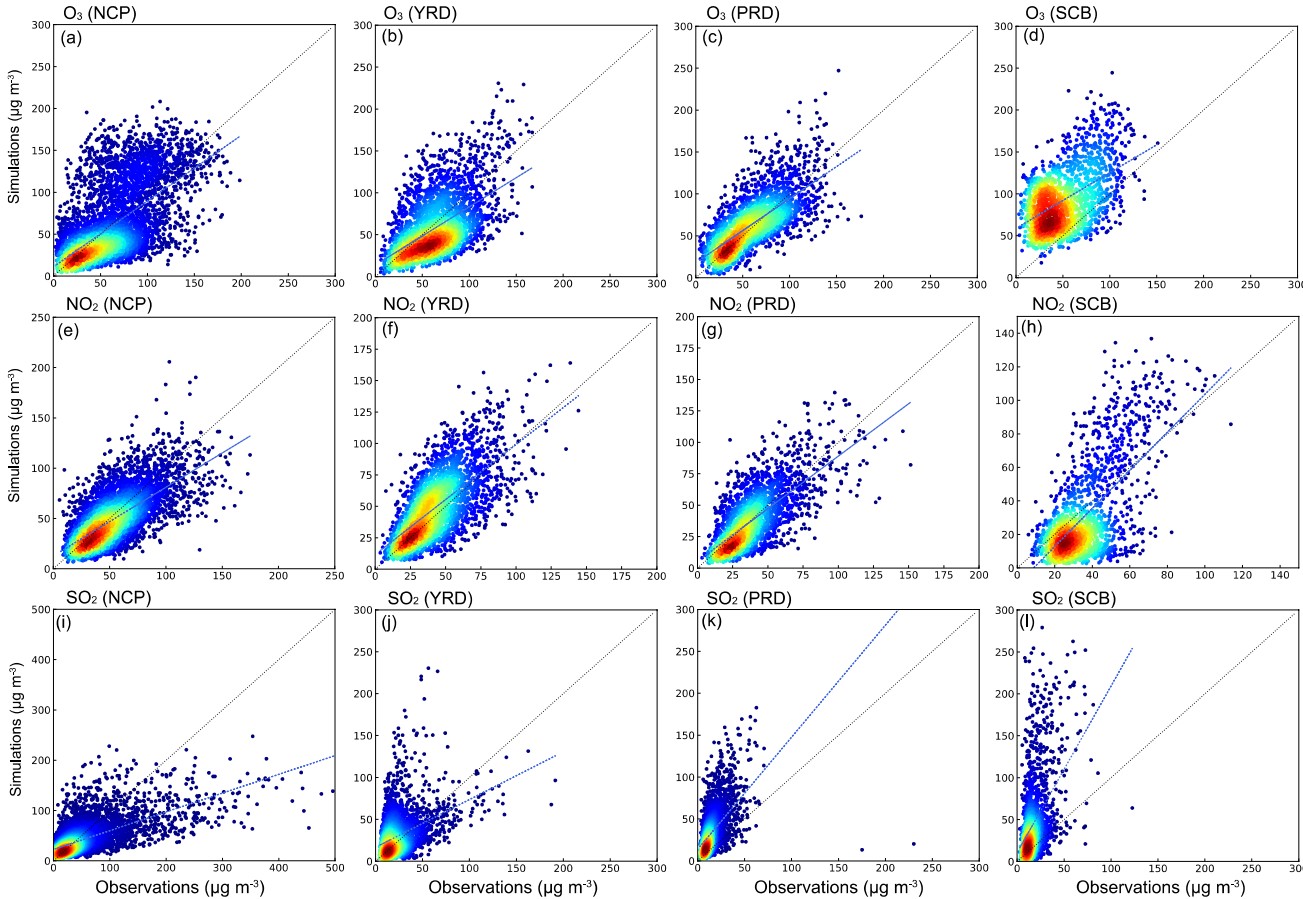

**Figure 4. Scatter plots of modelled and observed hourly concentrations of (a-d) O₃, (e-h) NO₂ and (i-l) SO₂ in the NCP, YRD, PRD, and SCB regions.**

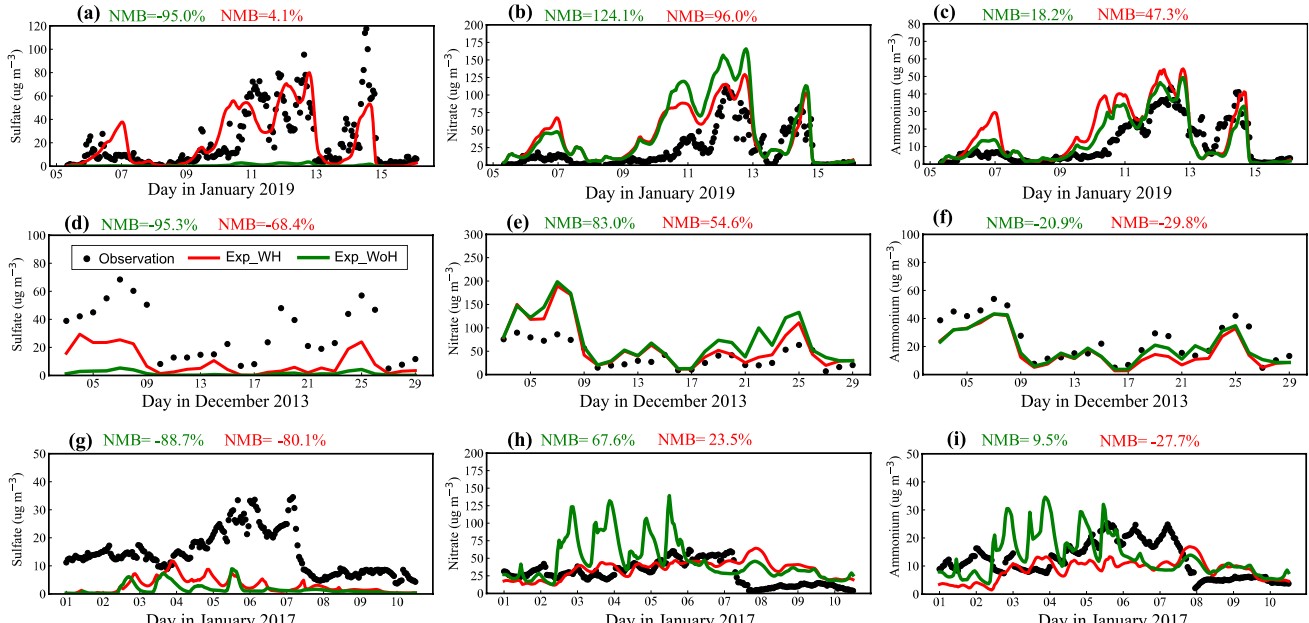

**Figure 5. Observed and simulated hourly SIA concentrations from the Exp_WH and Exp_WoH experiments at the (a-c) Langfang, (d-f) Nanjing and (g-i) Chengdu sites.**

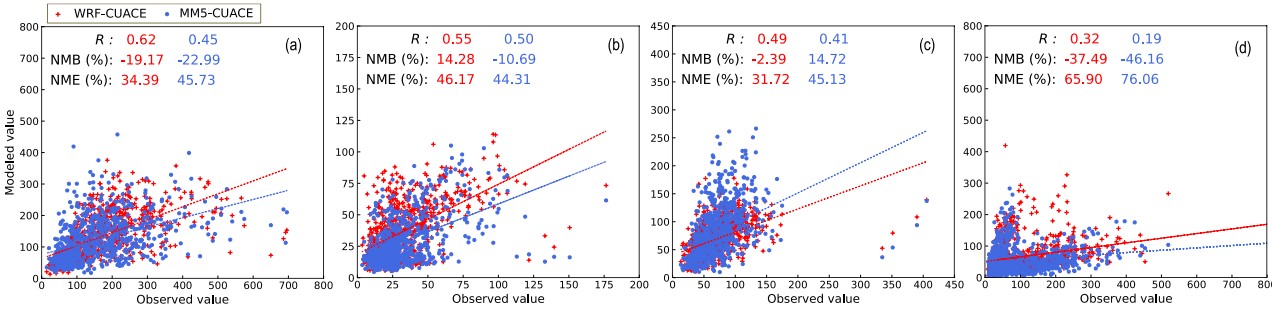

**Figure 6. Scatter plots of simulated, with (blue) MM5/CUACE and (red) WRF/CUACE, and observed daily concentrations of (a) PM$_{2.5}$, (b) O$_3$, (c) NO$_2$ and (d) SO$_2$.**

**Table 1 Uptake coefficients for reactions (14)-(22).**

| Gas species | Uptake coefficients | References |
|---|---|---|
| $H_2O_2$ | $\gamma = 1.0 \times 10^{-4}$ | Bian and Zender (2003) |
| $HNO_3$ | $\gamma = 1.0 \times 10^{-1}$ | Seisel et al. (2004) |
| $HO_2$ | $\gamma = 1.0 \times 10^{-1}$ | Phadnis and Carmichael (2000) |
| $N_2O_5$ | $\gamma = \begin{cases} \gamma_{low}, RH \in [0,50\%] \\ \gamma_{low} + \dfrac{(\gamma_{high} - \gamma_{low})}{(RH_{max} - 0.5)} * (RH - 0.5), RH \in (50\%, RH_{max}] \\ \gamma_{high}, RH \in (RH_{max}, 100\%] \end{cases}$ | Wang et al. (2012) Zheng et al. (2015) |
| $NO_2$ | | |
| $NO_3$ | | |
| $SO_2$ | | |
| $O_3$ | $\gamma = 3.0 \times 10^{-5}$ | Michel et al. (2003) |
| OH | $\gamma = 1.0 \times 10^{-4}$ | Zhang and Carmichael (1999) |

* The $\gamma_{low}$ and $\gamma_{high}$ are the lower and upper limits of $\gamma$ values. The $RH_{max}$ is the $RH$ value at which the $\gamma$ reaches the upper limit. The values of $\gamma_{low}$, $\gamma_{high}$ and $RH_{max}$ are referred to the work of Zheng et al. (2015) and Wang et al., (2012). That is, values of $\gamma_{low}$ for $N_2O_5$, $NO_2$, $NO_3$ and $SO_2$ are 1E-3, 4.4E-5, 0.1 and 2E-5, respectively corresponding to the values of $\gamma_{high}$ at 0.1, 2E-4, 0.23, 5E-5. The $RH_{max}$ is 70 % for $N_xO_y$, and is 100 % for $SO_2$.

**Table 2 Physical parameterization schemes used in the modelling study.**

| Physical management | Parameterization | References |
|---|---|---|
| Microphysics scheme | Lin | Lin et al. (1983) |
| Shortwave radiation | Goddard | Chou and Suarez (1994) |
| Longwage radiation | RRTM | Mlawer et al. (1997) |
| Land surface scheme | Noah | Chen and Dudhia (2001) |
| Boundary layer scheme | MYJ | Janjić (1994) |
| Cumulus scheme | Grell-3D | Grell (1993) |

**Table 3 Statistical metrics for hourly PM$_{2.5}$ in four haze contaminated areas (2013–2017), in which bold, normal , and italic font for MFB and MFE correspond to the "excellent", "good", and "average" levels in Morris et al. (2005), respectively.**

| | R | MB | ME | NMB | NME | MFB | MFE |
|---|---|---|---|---|---|---|---|
| | | μg m$^{-3}$ | μg m$^{-3}$ | % | % | % | % |
| **NCP** | 0.59 | -5.0 | 44.5 | -5.4 | 47.5 | 3.3 | 49.1 |
| Winter | 0.59 | -45.0 | 67.7 | -28.4 | 42.7 | -22.5 | 47.0 |
| Spring | 0.57 | -9.5 | 28.0 | -14.0 | 41.1 | -20.7 | 47.4 |
| Summer | 0.47 | 33.9 | 42.9 | 55.1 | 69.8 | *44.9* | *56.3* |
| Autumn | 0.63 | -0.8 | 39.2 | -0.9 | 45.4 | 9.0 | 45.9 |
| **YRD** | 0.71 | 12.9 | 26.9 | 21.8 | 45.3 | 21.1 | 42.9 |
| Winter | 0.75 | 6.0 | 30.6 | 6.4 | 32.5 | **8.5** | **34.1** |
| Spring | 0.49 | 14.2 | 26.3 | 25.4 | 47.1 | 19.1 | 40.0 |
| Summer | 0.56 | 16.4 | 23.3 | 47.8 | 67.9 | 26.7 | 49.4 |
| Autumn | 0.66 | 15.1 | 27.3 | 28.7 | 51.8 | 29.5 | 48.0 |
| **PRD** | 0.68 | 5.3 | 17.1 | 13.1 | 42.1 | 8.6 | 40.1 |
| Winter | 0.56 | 3.0 | 20.5 | 5.0 | 34.6 | **5.5** | **34.4** |
| Spring | 0.64 | 6.9 | 17.6 | 19.5 | 49.7 | 4.2 | 45.6 |
| Summer | 0.68 | 2.8 | 8.5 | 14.8 | 44.4 | 5.9 | 39.0 |
| Autumn | 0.54 | 8.6 | 21.8 | 17.7 | 45.2 | 18.3 | 41.9 |
| **SCB** | 0.59 | 7.6 | 31.3 | 12.2 | 50.3 | *20.7* | *51.4* |
| Winter | 0.41 | -13.3 | 46.7 | -11.5 | 40.4 | -8.3 | 45.2 |
| Spring | 0.49 | 4.1 | 22.4 | 8.4 | 45.9 | 11.4 | 46.1 |
| Summer | 0.40 | 21.6 | 28.2 | 60.4 | 78.6 | *38.7* | *58.9* |
| Autumn | 0.58 | 15.9 | 28.2 | 31.4 | 55.7 | *37.2* | *54.3* |

**Table 4 Statistical metrics for $O_3$ and $NO_2$ concentrations. Criteria for $O_3$ are from the EPA (2005, 2007). The values that do not meet the criteria are in bold.**

| Variables | | NCP | YRD | PRD | SCB | Criteria |
|---|---|---|---|---|---|---|
| $O_3$ | R | 0.64 | 0.66 | 0.77 | 0.60 | |
| | NMB (%) | -0.60 | -8.21 | 7.24 | **77.61** | $\leqslant\pm15$ |
| | IOA | 0.80 | 0.80 | 0.87 | 0.67 | |
| $NO_2$ | R | 0.60 | 0.64 | 0.67 | 0.57 | |
| | NMB (%) | -6.62 | 14.42 | -2.45 | -14.36 | |
| | IOA | 0.77 | 0.77 | 0.81 | 0.71 | |
| $SO_2$ | R | 0.65 | 0.41 | 0.57 | 0.47 | |
| | NMB (%) | -15.48 | 24.55 | 125.74 | 159.44 | |
| | IOA | 0.72 | 0.60 | 0.49 | 0.32 | |