# Peer review of "Development of WRF/CUACE v1.0 model and its preliminary application in simulating air quality in China"

_Geoscientific Model Development, 2020_

## Referee Comment (RC1) · Anonymous Referee #1 · 31 Jul 2020

In this manuscript the authors updated the CUACE model with heterogenous reactions and updated dry deposition scheme of particles, and coupled it to the WRF model. This study also evaluated the WRF/CUACE v1.0 model by simulating PM2.5, O3, and NO2 concentrations in different seasons and different years. This article is clearly written and the methods are generally sound. I recommend the manuscript to be published unless the following comments are addressed:

1. Line 234-235ïïjŽThe authors mentioned "The feedback of chemical species on meteorology in the current WRF/CUACE version is not realized". So in Figure 1, I suggest using dashed line to indicate the influence of chemical variables on WRF module. 2.

Line 290-291: The simulations are relatively poor in the SCB, where the complex terrain poses great challenges to meteorological field simulations. Show the simulations results of the meteorological fields of the four regions in the supplementary, and compare the simulation results with in-situ observations. 3. In Section 5.3, the authors evaluated the model performance with and without heterogeneous chemical reactions during a haze event at the Langfang site. How about model improvements at the other sites in the YRD, PRD and SCB region? 4. Line 90-91: This study also updated the dry deposition scheme of particles in CUACE. Please also show the model improvements with and without the updated dry deposition scheme in the supplementary.

---

## Referee Comment (RC2) · Anonymous Referee #2 · 10 Aug 2020

**Review**

**"Development of WRF/CUACE v1.0 model and its preliminary application in simulating air quality in China" by Lei Zhang et al.**

This publication presents a new model called "WRF/CUACE" being the implementation of the chemistry model CUACE into the NWP model WRF version 3. This new model is similar in his implementation to WRF-chem. The authors also presents new developments on aerosol dry deposition scheme and heterogeneous chemistry. The model is evaluated over China on several selected month and deals with PM2.5, ozone and $NO_2$. An other evaluation deals with the model ability to simulate secondary inorganic aerosols and shows the impact of heterogeneous chemistry freshly developed.

This publication is interesting as it presents a new model and proves the feasibility of an easy implementation of a chemistry module into WRF-Chem. But the description of the different compounds are not precise enough and some references are lacking. The available code is very hard to navigate and to understand what part is used, especially concerning the chemical scheme. I just navigate in the directories without trying to compile and run it.

**General comments**

This new model aims at replace the actual operational coupled model CUACE with MM5/GRAPES, because the development of the MM5 model has been stopped in favour of WRF. There are no comparison between the actual model and the new WRF/CUACE model. Yet it might have been interesting to compare these two model in order to assess the viability of the newly developed model.

It is not very clear how the different processes are treated by the different sub-model. For example at page 4 on line 108: "emissions, gaseous chemistry, and a size-segregated multicomponent aerosol algorithm (Zhou et al., 2012), and has been designed as a unified chemistry module". But on line 130 the authors said CUACE also treat particle dry deposition. The authors need to clarify what processes is done by which model. This includes the Figure 1 where it would be interesting to have a CUACE box that shows what in included in CUACE. Also on Figure 1 processes done by WRF need to be in the WRF box (convection for example). Also consider to rewrite the section 4, as a reader does not necessarily know how the model WRF-Chem works.

In section 3.2, the authors describe the added heterogeneous chemistry added to the model. I wonder if "Aerosol" stands for all the aerosols in the model, treated the same way or if only a sample of all aerosols are considered in the reaction. Also, the way the reactions are written may let think that the aerosol used as a reactant disappear, or I guess it only acts as a support for the reaction.

The description of the model CUACE is not precise enough, essentially concerning the chemical scheme and the reference Zhou et al, (2012) does not either. You claim that RADM2 has 121 reactions, but there are more in Stockwell et al, (1990). Please add the reference for RADM2

and explain the differences between the original publication and you version of RADM2. In section 4, authors explain they added the possibility to use the chemical scheme CBM-Z using KPP. But they do not precise which chemical scheme is finally used. If it is RADM2, then this section should be in the conclusion as future work. If it is CBM-Z then it should be on section 2.2 about CUACE module and more developed: number of species, number of reactions, number of photochemical reactions, way the photochemical reactions are taken into account (especially above the 100hPa upper limit), etc.

The present paper deals with a new combination of a NWP and a chemistry model. But only a part of the chemistry is evaluated. It would have been interesting to evaluate the meteorological fields during the simulation made. Moreover the fact that the SCB region seems badly represented for PM2.5 is due to the complex terrain could be illustrated.

In section 5.2, the authors talk about the negative bias in winter in NCP region by saying that the model misses secondary aerosols. But in summer it seems to be a positive bias almost as dramatic as the negative bias in winter. Do the authors have an explanation for this bias?

The authors detailed the implementation of the new dry deposition scheme. Also in the conclusion, they wrote "it is difficult to evaluate he dry deposition process is improved", but they did not present any comparison between the two parametrization. A comparison over the already used observed concentrations for the evaluation might be a start for evaluating the improvement.

**Specific comments**

- Page 3, line 70: A or several reference for WRF are missing here.

- Page 4, line 114: Please add 'primary' for organic carbon if it is the case. Otherwise add a sentence to explain how secondary organic aerosols are treated.

- Page 4, line 123: Please add the fact that $X_i$ is the mixing ratio of the species i.

- Page 4, line 124: I do not understand what the authors mean by clear-air tendency, please explain.

- Page 5/6: Generally speaking this part on deposition is not always easy to read because the are parenthesis missing for function [e.g. tanhη → tan(hη)] or multiply sign also missing (e.g. LAIE$_T$h → LAI*E$_T$*h).

- Page 5, line 132: "that developed by Petroff and Zhang" → " developed by Petroff and Zhang" for example.

- page 5, line 138: Please add a sentence saying that $V_d$ is the dry deposition velocity.

- Page 5, line 143: $V_g$ and $V_{phor}$ are not detailed. Please add a formula or a reference for both of them.

- Page 5, line 153: It is not clear that $E_g = E_{gb} + E_{gt.}$

- Page 5, line 159: $t_{ph}^+$ is not detailed. Please add a reference or a formula.

- Page 5, line 183: $R_s$ is not defined.

- Page 7, line 216: What is "chem_opt(122)"?

- Page 8, line 223: A reference is missing for KPP.

- Page 8, line 247: The authors does not specify whether WRF is used in hydrostatic or NH mode.

- Page 9, line 268: Is it possible to add a figure showing the extent of the MEIC inventory? Maybe it could be added on Figure 2.

- Page 9, line 270: Why do the authors use anthropogenic emissions representative for 2012, 2014 and 2016 to represent the years 2013, 2015 and 2017? Moreover for which year(s) is the MIX inventory representative?

- Page 10, line 296: Please add the mention 'not shown' for the time series comparison.

- Page 10, line 303: Please add a reference for the aerosol composition.

- Page 11, line 351: Please explain what is the index of agreement exactly.

- Page 11, line 351: Why do the authors only evaluate the simulations against $O_3$ and $NO_2$ observations? Indeed $SO_2$ observations might be a good observation since it is the direct precursor for sulfate aerosols.

- Figure 3: (a), (b), (c) and (d) are missing on the figure. The 3 of mg m$^{-3}$ is not in exponent size.

- Table 1: What are the value of $\gamma_{low}$ and $\gamma_{high}$? What is the value of $RH_{max}$? There seems to be a problem at the end of the line with a lonely bracket for the uptake coefficient for $N_xO_y$ and $SO_2$.

- Table 3: Please add "hourly" in the description of the table.

---

## Author Comment (AC1) · 25 Sep 2020

**Dear editor and referee#1,**

Thank you very much for your time and attentions on this work. The comments and suggestions are very useful to improve our manuscript. Following is a point-by-point response to referee #1's comments. Texts in italic are the comments, those in black bold are our responses. We hope that you will find the changes satisfactory.

*In this manuscript the authors updated the CUACE model with heterogenous reactions and updated dry deposition scheme of particles, and coupled it to the WRF model. This study also evaluated the WRF/CUACE v1.0 model by simulating $PM_{2.5}$, $O_3$, and $NO_2$ concentrations in different seasons and different years. This article is clearly written and the methods are generally sound. I recommend the manuscript to be published unless the following comments are addressed:*

*1. Line 234-235: The authors mentioned "The feedback of chemical species on meteorology in the current WRF/CUACE version is not realized". So in Figure 1, I suggest using dashed line to indicate the influence of chemical variables on WRF module.*

**Response: Thanks for pointing it out. It has been modified to dashed line in the revised manuscript.**

*2.Line 290-291: The simulations are relatively poor in the SCB, where the complex terrain poses great challenges to meteorological field simulations. Show the simulations results of the meteorological fields of the four regions in the supplementary, and compare the simulation results with in-situ observations.*

**Response: Following the suggestion, the simulations results of the meteorological fields of the four regions were added in the supplementary (as shown in Table R1). It can be seen that the simulations of meteorological fields in the SCB are relatively**

poor than the other three regions. For example, the *R*, MB, NMB and RMSE values of T2 in the SCB are 0.88, 1.52 °C, 9.95 % and 2.50 °C, respectively, while the values in the other three regions vary from 0.91 to 0.93, 0.48 to 1.14 °C, 5.31 to 7.01 % and 2.01 to 2.39 °C. The *R* value of WS10 in the SCB is 0.40, which is obviously worse than that of the other three regions (ranging from 0.60 to 0.74), indicating the variation of WS10 in the SCB was not well reproduced by the model. We have added the comparison in Section 5.2 in the revised manuscript.

**Table R1** Statistical metrics for hourly temperature at 2 m (T2), hourly relative humidity at 2 m (RH2) and hourly wind speed at 10 m (WS10), respectively in the NCP, YRD, PRD and SCB regions.

| | | Obs | Sim | *R* | MB | ME | NMB | RMSE |
|---|---|---|---|---|---|---|---|---|
| **NCP** | T2 (°C) | 17.31 | 18.07 | 0.91 | 0.76 | 1.87 | 7.01 % | 2.34 |
| | RH2 (%) | 62.88 | 51.10 | 0.80 | -11.78 | 14.47 | -18.94 % | 17.91 |
| | WS10 (m/s) | 2.05 | 2.99 | 0.64 | 0.95 | 1.29 | 52.40 % | 1.60 |
| **YRD** | T2 (°C) | 17.29 | 17.77 | 0.93 | 0.48 | 1.62 | 6.34 % | 2.01 |
| | RH2 (%) | 70.74 | 64.51 | 0.82 | -6.22 | 11.28 | -8.55 % | 13.95 |
| | WS10 (m/s) | 2.42 | 3.29 | 0.74 | 0.87 | 1.20 | 39.75 % | 1.47 |
| **PRD** | T2 (°C) | 22.92 | 24.06 | 0.91 | 1.14 | 2.06 | 5.31 % | 2.39 |
| | RH2 (%) | 75.74 | 67.20 | 0.78 | -8.54 | 12.73 | -10.72 % | 14.88 |
| | WS10 (m/s) | 2.23 | 3.23 | 0.60 | 1.01 | 1.32 | 48.73 % | 1.61 |
| **SCB** | T2 (°C) | 18.02 | 19.53 | 0.88 | 1.52 | 2.04 | 9.95 % | 2.50 |
| | RH2 (%) | 74.17 | 59.87 | 0.73 | -14.30 | 15.98 | -19.00 % | 18.77 |
| | WS10 (m/s) | 1.35 | 2.05 | 0.40 | 0.70 | 0.99 | 60.26 % | 1.24 |

\* All *R* (correlation coefficient) values passed $p < 0.001$.

\* Obs and Sim represent the average observations and simulations, respectively.

*3. In Section 5.3, the authors evaluated the model performance with and without heterogeneous chemical reactions during a haze event at the Langfang site. How about model improvements at the other sites in the YRD, PRD and SCB region?*

**Response: Sincere thanks for the suggestions. We have tried our best to collect observations of inorganic secondary aerosols in the three regions. So far, the observations from 3 to 29 December 2013 in Nanjing (located in the YRD) and from 1 to 10 January 2017 in Chengdu (located in the SCB) are obtained for evaluation (Fig. R1). As shown in Fig. R1, simulations of sulfate and nitrate in the two sites**

are generally improved (change in bias from −95.3 % to -68.4 % in Nanjing and from -88.7 % to -80.1 % in Chengdu for sulfate; change in bias from 83.0 % to 54.6 % in Nanjing and from 67.6 % to 23.5 % in Chengdu for nitrate). The results were added in Section 5.3 in the revised manuscript. We will continue to collect data in the PRD for evaluation in future work.

[Figure]

**Figure R1.** Observed and simulated hourly SIA concentrations from the Exp_WH and Exp_WoH experiments at the (a-c) Nanjing and (d-f) Chengdu site.

*4. Line 90-91: This study also updated the dry deposition scheme of particles in CUACE. Please also show the model improvements with and without the updated dry deposition scheme in the supplementary.*

**Response: Thanks very much for the suggestions. We performed simulations for a winter month (January in 2015) to show the model improvements with and without the updated dry deposition scheme. As shown in Fig. R2, the PM$_{2.5}$ concentrations were commonly underestimated with the Z01 scheme (Fig. R2a), as it tends to overestimate the dry deposition velocity of fine particles (Petroff and Zhang, 2010). The underestimation was improved significantly when the Z01 scheme was updated to the PZ10 scheme (Fig. R2b). We have added the improvements in the supplementary.**

[Figure]

**Figure R2.** Observed and simulated PM$_{2.5}$ concentrations with (a) Z01 and (b) PZ10 particle dry deposition schemes.

**Reference:**

Petroff, A. and Zhang, L.: Development and validation of a size-resolved particle dry deposition scheme for application in aerosol transport models, Geoscientific Model Development, 3, 753-769, 2010.

---

## Author Comment (AC2) · 27 Sep 2020

**Dear editor and referee#2,**

Thank you very much for your time and attentions on this work. The comments and suggestions are very useful to improve our manuscript. Following is a point-by-point response to referee #2's comments. Texts in italic are the comments, those in black bold are our responses. We hope that you will find the changes satisfactory.

This publication presents a new model called "WRF/CUACE" being the implementation of the chemistry model CUACE into the NWP model WRF version 3. This new model is similar in his implementation to WRF-chem. The authors also presents new developments on aerosol dry deposition scheme and heterogeneous chemistry. The model is evaluated over China on several selected month and deals with PM2.5, ozone and NO2. An other evaluation deals with the model ability to simulate secondary inorganic aerosols and shows the impact of heterogeneous chemistry freshly developed.

This publication is interesting as it presents a new model and proves the feasibility of an easy implementation of a chemistry module into WRF-Chem. But the description of the different compounds are not precise enough and some references are lacking. The available code is very hard to navigate and to understand what part is used, especially concerning the chemical scheme. I just navigate in the directories without trying to compile and run it.

**General comments:**

This new model aims at replace the actual operational coupled model CUACE with MM5/GRAPES, because the development of the MM5 model has been stopped in favour of WRF. There are no comparison between the actual model and the new WRF/CUACE model. Yet it might have been interesting to compare these two model in order to assess

**the viability of the newly developed model.**

Response: Thanks for this suggestion. We agree with the reviewer on the need to compare the newly developed model with MM5/CUACE or GRAPES/CUACE model. To this end, we have now obtained  $PM_{2.5}$  concentrations in December 2013 simulated by Jiang et al. (2015) using the GRAPES/CUACE model. As the model domain setting, anthropogenic emission inventory (MEIC2012) and reanalysis data (NCEP-FNL) used in Jiang et al., (2015) are generally same to those in our study, so the comparison results are convincing. As shown in Fig. R1, the biases of the GRAPES/CUACE and WRF/CUACE model exhibit no significant difference. However, the correlation coefficients (R) of WRF/CUACE simulation are commonly higher than those of GRAPES/CUACE simulation. It is known that daily variation of air pollutants are generally driven by change of meteorology, indicating the meteorology simulation by the WRF are better than the GRAPES. We have added the above analysis in Section 5.2.3 in the revised manuscript.

---

## Author Comment (AC3) · 28 Sep 2020

**Dear editor and referee#2,**

Thank you very much for your time and attentions on this work. The comments and suggestions are very useful to improve our manuscript. Following is a point-by-point response to referee #2's comments. Texts in italic are the comments, those in black bold are our responses. We hope that you will find the changes satisfactory.

*This publication presents a new model called "WRF/CUACE" being the implementation of the chemistry model CUACE into the NWP model WRF version 3. This new model is similar in his implementation to WRF-chem. The authors also presents new developments on aerosol dry deposition scheme and heterogeneous chemistry. The model is evaluated over China on several selected month and deals with $PM_{2.5}$, ozone and NO2. An other evaluation deals with the model ability to simulate secondary inorganic aerosols and shows the impact of heterogeneous chemistry freshly developed.*

*This publication is interesting as it presents a new model and proves the feasibility of an easy implementation of a chemistry module into WRF-Chem. But the description of the different compounds are not precise enough and some references are lacking. The available code is very hard to navigate and to understand what part is used, especially concerning the chemical scheme. I just navigate in the directories without trying to compile and run it.*

*General comments:*

*This new model aims at replace the actual operational coupled model CUACE with MM5/GRAPES, because the development of the MM5 model has been stopped in favour of WRF. There are no comparison between the actual model and the new WRF/CUACE model. Yet it might have been interesting to compare these two model in order to assess*

*the viability of the newly developed model.*

**Response: Thanks for this suggestion. We agree with the reviewer on the need to compare the newly developed model with MM5/CUACE or GRAPES/CUACE model. To this end, we have now obtained PM$_{2.5}$ concentrations in December 2013 simulated by Jiang et al. (2015) using the GRAPES/CUACE model. As the model domain setting, anthropogenic emission inventory (MEIC2012) and reanalysis data (NCEP-FNL) used in Jiang et al., (2015) are generally same to those in our study, so the comparison results are convincing. As shown in Fig. R1, the biases of the GRAPES/CUACE and WRF/CUACE model exhibit no significant difference. However, the correlation coefficients (*R*) of WRF/CUACE simulation are commonly higher than those of GRAPES/CUACE simulation. It is known that daily variation of air pollutants are generally driven by change of meteorology, indicating the meteorology simulation by the WRF are better than the GRAPES. We have added the above analysis in Section 5.2.3 in the revised manuscript.**

[Figure]

**Figure R1.** Daily variations of PM$_{2.5}$ concentrations from observation (black solid

circles), simulation by GRAPES-CUACE (blue lines) and simulation by WRF/CUACE v1.0 (red lines).

**Reference:**

Jiang, C., Wang, H., Zhao, T., Li, T., and Che, H.: Modeling study of PM2.5 pollutant transport across cities in China's Jing-JinJi region during a severe haze episode in December 2013, Atmos. Chem. Phys., 15, 5803–5814, https://doi.org/10.5194/acp15-5803-2015, 2015.

*It is not very clear how the different processes are treated by the different sub-model. For example at page 4 on line 108: "emissions, gaseous chemistry, and a size-segregated multicomponent aerosol algorithm (Zhou et al., 2012), and has been designed as a unified chemistry module". But on line 130 the authors said CUACE also treat particle dry deposition. The authors need to clarify what processes is done by which model. This includes the Figure 1 where it would be interesting to have a CUACE box that shows what in included in CUACE. Also on Figure 1 processes done by WRF need to be in the WRF box (convection for example). Also consider to rewrite the section 4, as a reader does not necessarily know how the model WRF-Chem works.*

**Response: Sincere thanks for pointing this out. We have revised the Figure 1 to clearly describe the different processes treated in the different sub-model. As the CUACE is coupled with WRF based on the framework of WRF-Chem and shares the emission, vertical mixing and gas dry deposition scheme, we also include the processes of WRF-Chem in Figure 1.**

**Sorry for the unclear description. The CUACE is a unified chemistry module, in which most of the physical and chemical processes are treated (Fig. 1), except transport that generally done by host model (i.e., WRF). In the manuscript, the sentence "*The CUACE module is a unified atmospheric chemistry module incorporating three major functional modules: emissions, gaseous chemistry, and a size-segregated multicomponent aerosol algorithm (Zhou et al., 2012), and has been designed as a unified chemistry module*" has been revised as "__The CUACE is a__**

**unified chemistry module, which treats most of the physical and chemical processes (Fig. 1), except transport that generally done by a host model (such as GRAPES). The main processes treated in CUACE include emissions, gas chemistry, dry/wet deposition, cloud chemistry, nucleation/condensation and coagulation."**

We have carefully revised the Section 4 for readers to more easily understand how the WRF-Chem model works. For example, we added more description of WRF-Chem: "**The WRF-Chem model is a meteorology-chemistry coupled models. In the chemical module of WRF-Chem, the processes are split to emissions, vertical mixing, dry deposition, convection, gas chemistry, cloud chemistry, aerosol chemistry and wet deposition, all of which are integrated in a interface procedure (chem_driver). The transport process is done in the WRF model. The WRF-Chem uses registry tools for automatic generation of application code. Chemistry variables, as well as control options of chemical processes are coded in WRFV3/Registry/registry.chem, which provides the conveniences for developers to add variables and options.**"

[Figure]

**Figure 1.** Schematic of modules in the WRF/CUACE v1.0 system. Processes that treated in the chemical module of WRF/Chem are included in the "Chem" box.

*In section 3.2, the authors describe the added heterogeneous chemistry added to the model. I wonder if "Aerosol" stands for all the aerosols in the model, treated the same way or if only a sample of all aerosols are considered in the reaction. Also, the way the reactions are written may let think that the aerosol used as a reactant disappear, or I guess it only acts as a support for the reaction.*

**Response: Sorry for the unclear descriptions. The "Aerosol" stands for all the aerosols in the model. Aerosol in the reactions only acts as a support. We state in the revised Section 3.2 "Aerosol in the heterogeneous chemical reactions stands for all the aerosols in the model", and rewritten the forms of heterogeneous chemical**

**reactions from "… + Aerosol → …" to "… $\xrightarrow{\text{Aerosol}}$ …".**

*The description of the model CUACE is not precise enough, essentially concerning the chemical scheme and the reference Zhou et al, (2012) does not either. You claim that RADM2 has 121 reactions, but there are more in Stockwell et al, (1990). Please add the reference for RADM2 and explain the differences between the original publication and your version of RADM2. In section 4, authors explain they added the possibility to use the chemical scheme CBM-Z using KPP. But they do not precise which chemical scheme is finally used. If it is RADM2, then this section should be in the conclusion as future work. If it is CBM-Z then it should be on section 2.2 about CUACE module and more developed: number of species, number of reactions, number of photochemical reactions, way the photochemical reactions are taken into account (especially above the 100hPa upper limit), etc.*

**Response: Thanks very much for pointing out the mistake. We have confirmed with Zhou and checked the code of CUACE. There are totally 136 chemical reactions and 21 photochemical reactions in the RADM2 scheme in CUACE model. We have corrected the mistake in the revised Section 2.2.**

The CBM-Z is finally used for simulation. The CBM-Z photochemical mechanism (Zaveri and Peters, 1999) contains 55 species, 114 reactions and 20 photochemical reactions. The photochemical reactions are as follows:

$NO_2 + hv = NO + O3P$
$NO_3 + hv = .89\ NO_2 + .89\ O3P + .11\ NO$
$HONO + hv = OH + NO$
$HNO_3 + hv = OH + NO_2$
$HNO_4 + hv = HO_2 + NO_2$
$N_2O_5 + hv = NO_2 + NO_3$
$O_3 + hv = O3P$
$O_3 + hv = O1D$
$H_2O_2 + hv = 2\ OH$
$HCHO + hv = 2\ HO_2 + CO$
$HCHO + hv = CO$
$CH3OOH + hv = HCHO + HO_2 + OH$

ETHOOH + hv = ALD2 + HO2 + OH
ALD2 + hv = CH3O2 + HO2 + CO
AONE + hv = C2O3 + CH3O2
MGLY + hv = C2O3 + CO + HO2
OPEN + hv = C2O3 + CO + HO2
ROOH + hv = OH + 0.4 XO2 + 0.74 AONE +
        0.3 ALD2 + 0.1 ETHP + 0.9 HO2 + 1.98 XPAR
ONIT + hv = NO2 + 0.41 XO2 + 0.74 AONE +
        0.3 ALD2 + 0.1 ETHP + 0.9 HO2+ 1.98 XPAR
ISOPRD + hv = 0.97 C2O3 + 0.33 HO2 +
        0.33 CO + 0.7 CH3O2 + 0.2 HCHO +
        0.07 ALD2 + 0.03 AONE

**As the atmosphere above 100 hPa are most in stratospheric, and the CBMZ mostly focus on the troposphere, so the reactions above 100hPa are not take into account.**

*The present paper deals with a new combination of a NWP and a chemistry model. But only a part of the chemistry is evaluated. It would have been interesting to evaluate the meteorological fields during the simulation made. Moreover the fact that the SCB region seems badly represented for PM2.5 is due to the complex terrain could be illustrated.*

**Response: Sincere thanks for the suggestions. The simulated temperature at 2 m (T2), relative humidity at 2 m (RH2) and wind speed at 10 m (WS10) were selected for evaluation. Table R1 shows the observation mean, simulation mean, correlation coefficient ($R$), MB, ME, NMB and RMSE of the meteorological fields in the NCP, YRD, PRD and SCB, respectively.**

**The MB, RMSE and NMB for T2 vary from 0.48 to 1.14 °C, from 2.01 to 2.50 °C and from 5.31 to 9.95 %, respectively, indicating surface temperatures are slightly overestimated in the four regions. The $R$ value for T2, ranging from 0.88 to 0.93, indicates the variation trends are well captured by the model. The model underestimates RH2 in the four regions with the MB ranging from -6.22 to -14.30 % and the RMSE ranging from 13.95 to 18.77 %, which are comparable with previous studies in China (Wang et al., 2014; Gao et al., 2016). The RMSE for WS10 in the**

**four regions vary from 1.47 to 1.61 m s$^{-1}$, fall within the "good" model performance criteria (little than 2 m s$^{-1}$) proposed by Emery et al. (2001). However, it should be noted that the *R* for WS10 in the SCB is relatively poor, indicating the variation trends were not well captured. Generally, the model performed best in the YRD, followed by the PRD and NCP, and performed worst in the SCB. We have added the evaluation in Section 5.2.1 in the revised manuscript.**

**Table R1** Statistical metrics for hourly temperature at 2 m (T2), hourly relative humidity at 2 m (RH2) and hourly wind speed at 10 m (WS10), respectively in the NCP, YRD, PRD and SCB regions.

|  |  | Obs | Sim | *R* | MB | ME | NMB | RMSE |
|---|---|---|---|---|---|---|---|---|
| **NCP** | T2 (℃) | 17.31 | 18.07 | 0.91 | 0.76 | 1.87 | 7.01 % | 2.34 |
|  | RH2 (%) | 62.88 | 51.10 | 0.80 | -11.78 | 14.47 | -18.94 % | 17.91 |
|  | WS10 (m s$^{-1}$) | 2.05 | 2.99 | 0.64 | 0.95 | 1.29 | 52.40 % | 1.60 |
| **YRD** | T2 (℃) | 17.29 | 17.77 | 0.93 | 0.48 | 1.62 | 6.34 % | 2.01 |
|  | RH2 (%) | 70.74 | 64.51 | 0.82 | -6.22 | 11.28 | -8.55 % | 13.95 |
|  | WS10 (m s$^{-1}$) | 2.42 | 3.29 | 0.74 | 0.87 | 1.20 | 39.75 % | 1.47 |
| **PRD** | T2 (℃) | 22.92 | 24.06 | 0.91 | 1.14 | 2.06 | 5.31 % | 2.39 |
|  | RH2 (%) | 75.74 | 67.20 | 0.78 | -8.54 | 12.73 | -10.72 % | 14.88 |
|  | WS10 (m s$^{-1}$) | 2.23 | 3.23 | 0.60 | 1.01 | 1.32 | 48.73 % | 1.61 |
| **SCB** | T2 (℃) | 18.02 | 19.53 | 0.88 | 1.52 | 2.04 | 9.95 % | 2.50 |
|  | RH2 (%) | 74.17 | 59.87 | 0.73 | -14.30 | 15.98 | -19.00 % | 18.77 |
|  | WS10 (m s$^{-1}$) | 1.35 | 2.05 | 0.40 | 0.70 | 0.99 | 60.26 % | 1.24 |

\* All *R* (correlation coefficient) values passed *p* < 0.001.

\* Obs and Sim represent the average observations and simulations, respectively.

**Reference:**

Wang, Y., Zhang, Q., Jiang, J., Zhou, W., Wang, B., He, K., Duan, F., Zhang, Q., Philip, S., and Xie, Y.: Enhanced sulfate formation during China's severe winter haze episode in January 2013 missing from current models, J. Geophys. Res.-Atmos., 119, 10425–10440, doi:10.1002/2013JD021426, 2014.

Gao, M., Carmichael, G. R., Wang, Y., Ji, D., Liu, Z., and Wang, Z.: Improving simulations of sulfate aerosols during winter haze over Northern China: the impacts of heterogeneous oxidation by NO2, Frontiers of Environmental Science & Engineering, 10, 2016.

Emery, C., Tai, E., and Yarwood, G.: Enhanced meteorological modeling and performance evaluation for two Texas ozone episodes, in: Prepared for the Texas Natural Resource Conservation Commission, ENVIRON International Corporation, Novato, CA, USA, 2001.

*In section 5.2, the authors talk about the negative bias in winter in NCP region by saying that the model misses secondary aerosols. But in summer it seems to be a positive bias almost as dramatic as the negative bias in winter. Do the authors have an explanation for this bias?*

**Response: Thanks for pointing out it. According to our analysis, the positive bias in summer in NCP is mainly due to the uncertainty in anthropogenic emissions. PM₂.₅ concentration is mainly driven by primary emissions (internal cause) and meteorology (external cause). As shown in Table R2, there are not significant difference between the simulation of meteorology in winter and summer. We can also see that the dramatic difference in bias didn't happen in the YRD and PRD (Table R3), where the uncertainties of anthropogenic emissions are generally known as less than that of NCP, indicating significant uncertainties in the emission inventory in NCP. We have added the analysis in Section 5.2.2 in the revised manuscript.**

**Table R2** Statistical metrics for hourly temperature at 2 m (T2), hourly relative humidity at 2 m (RH2) and hourly wind speed at 10 m (WS10), respectively in winter and summer in the NCP.

| NCP region | | Obs | Sim | $R$ | MB | ME | NMB | RMSE |
|---|---|---|---|---|---|---|---|---|
| **Winter** | T2 (℃) | 1.59 | 2.01 | 0.85 | 0.42 | 1.67 | 34.4 % | 2.14 |
| | RH2 (%) | 62.65 | 53.17 | 0.75 | -9.48 | 14.18 | -15.62 % | 18.18 |
| | WS10 (m s⁻¹) | 1.82 | 2.64 | 0.62 | 0.82 | 1.15 | 50.17 % | 1.46 |
| **Summer** | T2 (℃) | 27.48 | 28.88 | 0.89 | 1.40 | 1.89 | 5.10 % | 2.38 |
| | RH2 (%) | 72.79 | 59.61 | 0.84 | -13.18 | 13.98 | -18.14 % | 16.42 |
| | WS10 (m s-1) | 1.93 | 2.42 | 0.54 | 0.49 | 1.00 | 31.3 % | 1.27 |

**Table R3** Statistical metrics for hourly PM₂.₅ in four haze contaminated areas (2013–2017), in which bold, normal , and italic font for MFB and MFE correspond to the "excellent", "good", and "average" levels in Morris et al. (2005), respectively.

| R | MB | ME | NMB | NME | MFB | MFE |
|---|---|---|---|---|---|---|
| | µg m⁻³ | µg m⁻³ | % | % | % | % |

| | | | | | | | |
|---|---|---|---|---|---|---|---|
| **NCP** | 0.59 | -5.0 | 44.5 | -5.4 | 47.5 | 3.3 | 49.1 |
| Winter | 0.59 | -45.0 | 67.7 | -28.4 | 42.7 | -22.5 | 47.0 |
| Spring | 0.57 | -9.5 | 28.0 | -14.0 | 41.1 | -20.7 | 47.4 |
| Summer | 0.47 | 33.9 | 42.9 | 55.1 | 69.8 | *44.9* | *56.3* |
| Autumn | 0.63 | -0.8 | 39.2 | -0.9 | 45.4 | 9.0 | 45.9 |
| **YRD** | 0.71 | 12.9 | 26.9 | 21.8 | 45.3 | 21.1 | 42.9 |
| Winter | 0.75 | 6.0 | 30.6 | 6.4 | 32.5 | **8.5** | **34.1** |
| Spring | 0.49 | 14.2 | 26.3 | 25.4 | 47.1 | 19.1 | 40.0 |
| Summer | 0.56 | 16.4 | 23.3 | 47.8 | 67.9 | 26.7 | 49.4 |
| Autumn | 0.66 | 15.1 | 27.3 | 28.7 | 51.8 | 29.5 | 48.0 |
| **PRD** | 0.68 | 5.3 | 17.1 | 13.1 | 42.1 | 8.6 | 40.1 |
| Winter | 0.56 | 3.0 | 20.5 | 5.0 | 34.6 | **5.5** | **34.4** |
| Spring | 0.64 | 6.9 | 17.6 | 19.5 | 49.7 | 4.2 | 45.6 |
| Summer | 0.68 | 2.8 | 8.5 | 14.8 | 44.4 | 5.9 | 39.0 |
| Autumn | 0.54 | 8.6 | 21.8 | 17.7 | 45.2 | 18.3 | 41.9 |
| **SCB** | 0.59 | 7.6 | 31.3 | 12.2 | 50.3 | *20.7* | *51.4* |
| Winter | 0.41 | -13.3 | 46.7 | -11.5 | 40.4 | -8.3 | 45.2 |
| Spring | 0.49 | 4.1 | 22.4 | 8.4 | 45.9 | 11.4 | 46.1 |
| Summer | 0.40 | 21.6 | 28.2 | 60.4 | 78.6 | *38.7* | *58.9* |
| Autumn | 0.58 | 15.9 | 28.2 | 31.4 | 55.7 | *37.2* | *54.3* |

*The authors detailed the implementation of the new dry deposition scheme. Also in the conclusion, they wrote "it is difficult to evaluate the dry deposition process is improved", but they did not present any comparison between the two parametrization. A comparison over the already used observed concentrations for the evaluation might be a start for evaluating the improvement.*

**Response: Thanks for the suggestions. We have added the equations and descriptions of Z01 scheme in the revised Section 3.1. The comparison between the two parametrization is added in the revised Section 3.1 "As previously revealed, Z01 scheme is tend to overestimate the dry deposition velocity of fine particles. The most significant difference between the Z01 and PZ10 scheme is the treatment of _Rs_, which stands for the dry velocity contributed by surface resistance, including the effect of Brownian diffusion, turbulent impaction, interception and rebound. According to the study of Wu et al., (2018), dry deposition velocity of fine particles**

**is strongly affected by the Brownian diffusion and turbulent impaction. Thereby, it could be inferred that the Z01 scheme is prone to overestimate the effect of Brownian diffusion and turbulent impaction"**.

Following the suggestions, we performed simulations for a winter month (January in 2015) to show the model improvements with and without the updated dry deposition scheme. As shown in Fig. R2, the PM$_{2.5}$ concentrations were commonly underestimated with the Z01 scheme (Fig. R2a), as it tends to overestimate the dry deposition velocity of fine particles (Petroff and Zhang, 2010). The underestimation was improved significantly when the Z01 scheme was updated to the PZ10 scheme (Fig. R2b). We have added the improvements in the supplementary.

[Figure]

**Figure R2.** Observed and simulated PM$_{2.5}$ concentrations with (a) Z01 and (b) PZ10 particle dry deposition schemes.

**Reference:**

Petroff, A. and Zhang, L.: Development and validation of a size-resolved particle dry deposition scheme for application in aerosol transport models, Geoscientific Model Development, 3, 753-769, 2010.

*Specific comments:*

*Page 3, line 70: A or several reference for WRF are missing here.*

**Response: Thanks for pointing out it. We have added the reference (Skamarock, 2008) in the revised manuscript.**

**Reference:**

Skamarock, W. C., Klemp, J. B., Dudhia, J., Gill, D. O., Barker, D. M., Duda, M. G., Huang, X.-Y., Wang, W., and Powers, J. G.: A description of the Advanced Research WRF version 3, National Center for Atmospheric Research Tech. Note, NCAR/TN-475+STR, 113 pp., 2008.

*Page 4, line 114: Please add 'primary' for organic carbon if it is the case. Otherwise add a sentence to explain how secondary organic aerosols are treated.*

**Response: The 'primary' was added.**

*Page 4, line 123: Please add the fact that Xi is the mixing ratio of the species i.*

**Response: It has been added.**

*Page 4, line 124: I do not understand what the authors mean by clear-air tendency, please explain.*

**Response: Thanks for pointing out it. The clear-air tendency means aerosol mass produced by chemical transformation of their precursors together with particle nucleation, condensation and coagulation form the clear-air processes (Gong et al., 2003). We have added the explanation in the revised manuscript.**

**Reference:**

Gong, S. L., Barrie, L. A., J.-P. Blanchet, Salzen, K. v., U. Lohmann, and Lesins, G.: Canadian Aerosol Module: A size-segregated simulation of atmospheric aerosol processes for climate and air quality models 1. Module development, Journal of Geophysical Research, 108, 2003.

*Page 5/6: Generally speaking this part on deposition is not always easy to read because there are parenthesis missing for function [e.g. tanhη → tan(hη)] or multiply sign also missing (e.g. LAIETh → LAI\*ET\*h).*

**Response: Thanks for pointing out it. We have carefully checked page 5/6. The parenthesis and multiply sign missed in this part has been added.**

*Page 5, line 132: "that developed by Petroff and Zhang" → " developed by Petroff and Zhang" for example.*

**Response: It has been deleted.**

*Page 5, line 138: Please add a sentence saying that Vd is the dry deposition velocity.*

**Response: The sentence has been added following the suggestion.**

*Page 5, line 143: Vg and Vphor are not detailed. Please add a formula or a reference for both of them.*

**Response: Thanks for pointing out it. Formulas for $V_g$ and $V_{phor}$ are added.**

*Page 5, line 153: It is not clear that Eg = Egb + Egt.*

**Response: Thanks for pointing out it. We have revised the description of Eg as Eg = Egb + Egt.**

*Page 5, line 159: tph+ is not detailed. Please add a reference or a formula.*

**Response: Thanks for pointing this out. The reference (Petroff et al., 2010) is added in the revised manuscript.**

**Reference:**

Petroff, A. and Zhang, L.: Development and validation of a size-resolved particle dry deposition scheme for application in aerosol transport models, Geoscientific Model Development, 3, 753-769, 2010.

*Page 5, line 183: Rs is not defined.*

**Response: Thanks for pointing out it. *Rs* is the surface resistance, which is generally expressed as the reciprocal of the surface deposition velocity ($V_{ds}$). It has been defined in the revised manuscript.**

*Page 7, line 216: What is "chem_opt(122)"?*

**Response: Thanks for pointing out it. The chem_opt is an option in WRF-Chem to choose which chemical scheme is used (e.g., 10 for CBMZ/MOSAIC). We add a option 122 for users to start the CUACE chemistry module. We have rewritten the description in the revised Section 4.**

*Page 8, line 223: A reference is missing for KPP.*

**Response: The reference (Damian et al., 2002) has been added following your suggestion.**

**Reference:**

Damian V, Sandu A, Damian M, Potra F, Carmichael G R. The kinetic preprocessor KPP-a software environment for solving chemical kinetics. Computers & Chemistry, 2002, 26(11): 1567–1579.

*Page 8, line 247: The authors does not specify whether WRF is used in hydrostatic or NH mode.*

**Response: Thanks for pointing out it. The WRF is used in NH mode. It has been specified in the revised Section 5.1.**

*Page 9, line 268: Is it possible to add a figure showing the extent of the MEIC inventory? Maybe it could be added on Figure 2.*

**Response: Thanks for the suggestion. In the revised manuscript, a new figure (Fig. 2) is added to show the extent of the MEIC inventory.**

[Figure]

**Figure 3.** Anthropogenic emissions of $PM_{2.5}$ (a,e,i), $NO_x$ (b,f,j), $SO_2$ (c,g,k), and CO (d,h,l), respectively in 2012 (a-d), 2014 (e-h), and 2016 (i-l). The unit is $\mu g\ m^{-2}\ s^{-1}$ for $PM_{2.5}$, $NO_x$, $SO_2$, and is $mg\ m^{-2}\ s^{-1}$ for CO.

*Page 9, line 270: Why do the authors use anthropogenic emissions representative for 2012, 2014 and 2016 to represent the years 2013, 2015 and 2017? Moreover for which year(s) is the MIX inventory representative?*

**Response: Currently, only the 2012, 2014 and 2016 MEIC inventory is open access for download. We use anthropogenic emissions representative for 2012, 2014 and 2016 to represent the years 2013, 2015 and 2017 in order to reflect the changes in anthropogenic emissions. The year of 2010 is the MIX inventory representative. We have added the above explanation in the revised Section 5.1.**

*Page 10, line 296: Please add the mention 'not shown' for the time series comparison.*

**Response: It has been added.**

*Page 10, line 303: Please add a reference for the aerosol composition.*

**Response: Following the suggestion, the reference (Huang et al., 2014) has been added.**

**Reference:**

Huang, R.J., Zhang, Y., Bozzetti, C., Ho, K.F., Cao, J.J., Han, Y., Daellenbach, K. R., Slowik, J. G., Platt, S. M., and Canonaco, F.: High secondary aerosol contribution to particulate pollution during haze events in China, Nature, 514, 218–222, 2014.

*Page 11, line 351: Please explain what is the index of agreement exactly.*

**Response: The index of agreement (IOA) is based on Willmott et al. (1980), which spans between 0 (indicating "complete disagreement") to 1 (indicating "complete agreement"). It is defined as equation (R1)**

$$\text{IOA} = 1 - \frac{\sum_{i=1}^{n}(P_i - O_i)^2}{\sum_{i=1}^{n}(|P_i - O| + |O_i - O|)} \ , \qquad\qquad (R1)$$

**where** *P, O and i represent simulation, observation and samples, respectively.* **The definition of IOA and the reference (Willmott et al., 1980) are added in the revised manuscript.**

**Reference:**

Willmott CJ, Wicks DE. 1980. An empirical method for the spatialinterpolation of monthly precipitation within California. Physical Geography 1: 59–73.

*Page 11, line 351: Why do the authors only evaluate the simulations against $O_3$ and $NO_2$ observations? Indeed $SO_2$ observations might be a good observation since it is the direct precursor for sulfate aerosols.*

**Response: Thanks very much for pointing out it. We have added the evaluation of $SO_2$ in Section 5.2.1 in the revised manuscript following the suggestion.**

*Figure 3: (a), (b), (c) and (d) are missing on the figure. The 3 of mg m-3 is not in exponent size.*

**Response: All are revised.**

*Table 1: What are the value of γlow and γhigh? What is the value of RHmax? There seems to be a problem at the end of the line with a lonely bracket for the uptake coefficient for $N_xO_y$ and $SO_2$.*

**Response: The $\gamma_{low}$ and $\gamma_{high}$ are the lower and upper limits of $\gamma$ values. The *$RH_{max}$* is the *RH* value at which the $\gamma$ reaches the upper limit. The values of $\gamma_{low}$, $\gamma_{high}$ and *$RH_{max}$* are referred to the work of Zheng et al. (2015). That is, values of $\gamma_{low}$ for $N_2O_5$, $NO_2$, $NO_3$ and $SO_2$ are 1E-3, 4.4E-5, 0.1 and 2E-5, respectively corresponding to the values of $\gamma_{high}$ at 0.1, 2E-4, 0.23, 5E-5. The *$RH_{max}$* is 70 % for $N_xO_y$, and is 100 % for $SO_2$. Thanks for pointing this out. We have added the description $\gamma_{low}$, $\gamma_{high}$ and *$RH_{max}$* in the revised Table.**

*Table 3: Please add "hourly" in the description of the table.*

**Response: It has been added.**

---

## Author Response (AR1)

**Dear Editor Remy and referees,**

**We are very grateful for your time and attentions on this work. Please find below our itemized responses to the referees' comments and a marked-up manuscript. We have addressed all the comments raised by both referees and incorporated them in the revised manuscript.**

**Thank you for your consideration.**

**Sincerely,**

**Sunling Gong, Tianliang Zhao, Lei Zhang et al.**
* * *
**Referee #1**

*In this manuscript the authors updated the CUACE model with heterogenous reactions and updated dry deposition scheme of particles, and coupled it to the WRF model. This study also evaluated the WRF/CUACE v1.0 model by simulating $PM_{2.5}$, $O_3$, and $NO_2$ concentrations in different seasons and different years. This article is clearly written and the methods are generally sound. I recommend the manuscript to be published unless the following comments are addressed:*

*1. Line 234-235: The authors mentioned "The feedback of chemical species on meteorology in the current WRF/CUACE version is not realized". So in Figure 1, I suggest using dashed line to indicate the influence of chemical variables on WRF module.*

**Response: Thanks for pointing it out. It has been modified to dashed line in the revised manuscript.**

*2.Line 290-291: The simulations are relatively poor in the SCB, where the complex terrain poses great challenges to meteorological field simulations. Show the simulations*

*results of the meteorological fields of the four regions in the supplementary, and compare the simulation results with in-situ observations.*

**Response: Sincere thanks for the suggestions. The the simulations results of the meteorological fields of the four regions were added in the supplementary (as shown in Table S1). It can be seen that the simulations of meteorological fields in the SCB are relatively poor than the other three regions. For example, the *R*, MB and RMSE values of T2 in the SCB are 0.88, 1.52 °C and 2.50 °C, respectively, while the values in the other three regions vary from 0.91 to 0.93, 0.48 to 1.14 °C and 2.01 to 2.39 °C. The *R* value of WS10 in the SCB is 0.40, which is obviously worse than that of the other three regions (ranging from 0.60 to 0.74), indicating the variation of WS10 in the SCB was not well reproduced by the model. We have added the comparison in Section 5.2.1 in the revised manuscript.**

Table S1 Statistical metrics for hourly temperature at 2 m (T2), hourly relative humidity at 2 m (RH2) and hourly wind speed at 10 m (WS10), respectively in the NCP, YRD, PRD and SCB regions.

|  |  | Obs | Sim | *R* | MB | ME | RMSE |
|---|---|---|---|---|---|---|---|
| **NCP** | T2 (°C) | 17.31 | 18.07 | 0.91 | 0.76 | 1.87 | 2.34 |
| | RH2 (%) | 62.88 | 51.10 | 0.80 | -11.78 | 14.47 | 17.91 |
| | WS10 (m s$^{-1}$) | 2.05 | 2.99 | 0.64 | 0.95 | 1.29 | 1.60 |
| **YRD** | T2 (°C) | 17.29 | 17.77 | 0.93 | 0.48 | 1.62 | 2.01 |
| | RH2 (%) | 70.74 | 64.51 | 0.82 | -6.22 | 11.28 | 13.95 |
| | WS10 (m s$^{-1}$) | 2.42 | 3.29 | 0.74 | 0.87 | 1.20 | 1.47 |
| **PRD** | T2 (°C) | 22.92 | 24.06 | 0.91 | 1.14 | 2.06 | 2.39 |
| | RH2 (%) | 75.74 | 67.20 | 0.78 | -8.54 | 12.73 | 14.88 |
| | WS10 (m s$^{-1}$) | 2.23 | 3.23 | 0.60 | 1.01 | 1.32 | 1.61 |
| **SCB** | T2 (°C) | 18.02 | 19.53 | 0.88 | 1.52 | 2.04 | 2.50 |
| | RH2 (%) | 74.17 | 59.87 | 0.73 | -14.30 | 15.98 | 18.77 |
| | WS10 (m s$^{-1}$) | 1.35 | 2.05 | 0.40 | 0.70 | 0.99 | 1.24 |

\* All *R* (correlation coefficient) values passed *p* < 0.001.

\* Obs and Sim represent the average observations and simulations, respectively.

*3. In Section 5.3, the authors evaluated the model performance with and without heterogeneous chemical reactions during a haze event at the Langfang site. How about model improvements at the other sites in the YRD, PRD and SCB region?*

**Response: Sincere thanks for the suggestions. We have tried our best to collect observations of inorganic secondary aerosols in the three regions. So far, the observations from 3 to 29 December 2013 in Nanjing (located in the YRD) and from 1 to 10 January 2017 in Chengdu (located in the SCB) are obtained for evaluation (Fig. R1). As shown in Fig. R1, simulations of sulfate and nitrate in the two sites are generally improved (change in bias from −95.3 % to -68.4 % in Nanjing and from -88.7 % to -80.1 % in Chengdu for sulfate; change in bias from 83.0 % to 54.6 % in Nanjing and from 67.6 % to 23.5 % in Chengdu for nitrate). The results were added in Section 5.3 in the revised manuscript. We will continue to collect data in the PRD for evaluation in future work.**

[Figure]

**Figure R1.** Observed and simulated hourly SIA concentrations from the Exp_WH and Exp_WoH experiments at the (a-c) Nanjing and (d-f) Chengdu site.

*4. Line 90-91: This study also updated the dry deposition scheme of particles in CUACE. Please also show the model improvements with and without the updated dry deposition scheme in the supplementary.*

**Response: Thanks very much for the suggestions. We performed simulations for a winter month (January in 2015) to show the model improvements with and without the updated dry deposition scheme. As shown in Fig. S3, the PM$_{2.5}$ concentrations were commonly underestimated with the Z01 scheme (Fig. R2a), as it tends to overestimate the dry deposition velocity of fine particles (Petroff and Zhang, 2010). The underestimation was improved significantly when the Z01 scheme was**

**updated to the PZ10 scheme (Fig. R2b). We have added the improvements in the supplementary.**

[Figure]

**Figure S3.** Observed and simulated PM$_{2.5}$ concentrations with (a) Z01 and (b) PZ10 particle dry deposition schemes.

**Reference:**

Petroff, A. and Zhang, L.: Development and validation of a size-resolved particle dry deposition scheme for application in aerosol transport models, Geoscientific Model Development, 3, 753-769, 2010.
* * *
**Referee #2**

*This publication presents a new model called "WRF/CUACE" being the implementation of the chemistry model CUACE into the NWP model WRF version 3. This new model is similar in his implementation to WRF-chem. The authors also presents new developments on aerosol dry deposition scheme and heterogeneous chemistry. The model is evaluated over China on several selected month and deals with PM$_{2.5}$, ozone and NO$_2$. An other evaluation deals with the model ability to simulate secondary inorganic aerosols and shows the impact of heterogeneous chemistry freshly developed.*

*This publication is interesting as it presents a new model and proves the feasibility of an easy implementation of a chemistry module into WRF-Chem. But the description of the different compounds is not precise enough and some references are lacking. The available code is very hard to navigate and to understand what part is used, especially concerning the chemical scheme. I just navigate in the directories without trying to compile and run it.*

90

**Response: We thank the reviewer for the valuable comments. The description of different compounds is modified to be more precise and the missing references are added. Please see our itemized responses below. To make it easier to navigate the code and to understand what part is used, a schematic showing the flow of information within WRF/CUACE is given in the supplementary (Figure S1).**

95

[Figure]

**Figure S1.** Schematic showing the flow of information within the WRF/CUACE v1.0 model. Gray

boxes indicate the newly added modules to the WRF/Chem framework to build the WRF/CUACE v1.0 model. Dashed boxes are descriptions of each module.

*General comments:*

*This new model aims at replace the actual operational coupled model CUACE with MM5/GRAPES, because the development of the MM5 model has been stopped in favour of WRF. There are no comparison between the actual model and the new WRF/CUACE model. Yet it might have been interesting to compare these two model in order to assess the viability of the newly developed model.*

**Response: Thanks for this suggestion. We agree with the reviewer on the need to compare the actual model and the new WRF/CUACE model. To this end, we performed a simulation using the MM5/CUACE for a winter month (January 2013), during which a long-lasting haze event occurred in central and eastern China. A new section (Section 5.4) is added in the revised manuscript:**

**"5.4 Comparison between the MM5/CUACE model and the WRF/CUACE v1.0 model**

**It is necessary to compare the MM5/CUACE model with the new WRF/CUACE model for the purpose of assessing the viability of the newly developed model. To this end, a simulation was performed using the MM5/CUACE model for a winter month, i.e., January 2013, during which a long-lasting haze event occurred in central and eastern China. The domain setting, anthropogenic emission inventory, initial and boundary fields of meteorology and chemistry are as the same as those of the WRF/CUACE in section 5.1. It should be known that the gas-phase chemistry mechanism and particle dry deposition scheme in MM5/CUACE model is RADM2 and Z01, respectively, that updated to CBM-Z and PZ10 in the new WRF/CUACE model. Physical parameterization used in the**

**MM5/CUACE is shown in Table S3 in the supplement.**

**Figure 6 presents a comparison of the modelled and observed daily concentrations of PM$_{2.5}$, O$_3$, NO$_2$ and SO$_2$ in the four regions. It can be seen that the concentrations of PM$_{2.5}$, NO$_2$ and SO$_2$ simulated in WRF/CUACE are closer to the observations relative to those of MM5/CUACE model (change in bias from -23.0 % to -19.2 % for PM$_{2.5}$, from 14.7 % to -2.4 % for NO$_2$ and from -46.2 % to -37.5 % for SO$_2$). The daily variations of the three species are also relatively better captured by the WRF/CUACE model (reflected by the *R* values changing from 0.45 to 0.62 for PM$_{2.5}$, from 0.41 to 0.49 for NO$_2$ and from 0.19 to 0.32 for SO$_2$). For O$_3$, the differences of statistical metrics between the two models are not obvious. The MM5/CUACE model performed with a slightly smaller bias of -10.7 % but with a lower R value of 0.50, which are 14.3 % and 0.55, respectively in the WRF/CUACE simulation. In summary, the new WRF/CUACE model performed better than the MM5/CUACE model in simulating air pollutants."**

[Figure]

**Figure 6.** Scatter plots of simulated, with (blue) MM5/CUACE and (red) WRF/CUACE, and observed daily concentrations of (a) PM$_{2.5}$, (b) O$_3$, (c) NO$_2$ and (d) SO$_2$.

*It is not very clear how the different processes are treated by the different sub-model. For example at page 4 on line 108: "emissions, gaseous chemistry, and a size-segregated multicomponent aerosol algorithm (Zhou et al., 2012), and has been designed as a unified chemistry module". But on line 130 the authors said CUACE also treat particle dry deposition. The authors need to clarify what processes is done by*

*which model. This includes the Figure 1 where it would be interesting to have a CUACE*
150 *box that shows what in included in CUACE. Also on Figure 1 processes done by WRF*
*need to be in the WRF box (convection for example). Also consider to rewrite the section*
*4, as a reader does not necessarily know how the model WRF-Chem works.*

**Response: Sincere thanks for pointing this out. The CUACE is a unified chemistry module, in which most of the physical and chemical processes are treated (Fig. 1),**
155 **except advection and convection processes that done by its host model. In the manuscript, the sentence "*The CUACE module is a unified atmospheric chemistry module incorporating three major functional modules: emissions, gaseous chemistry, and a size-segregated multicomponent aerosol algorithm (Zhou et al., 2012), and has been designed as a unified chemistry module*" has been revised as "The CUACE is**
160 **a unified chemistry module, which treats most of the physical and chemical processes, except advection and convection processes that done by its host model. The main processes treated in CUACE module include emissions, gas chemistry, dry and wet deposition, vertical mixing, aerosol-cloud interaction, and clear-air (i.e., aerosol produced by chemical transformation of their precursors together**
165 **with particle nucleation, condensation and coagulation) (An et al., 2016; Zhou et al., 2012; Gong et al., 2003).".**

**We have revised the Figure 1 and added Figure S1 to clearly describe the different processes treated in the different sub-model following your suggestion, and state in Section 4: "The flow of the major process splitting in the coupled WRF/CUACE**
170 **v1.0 model is illustrated in Fig. 1 with the structure of related subroutines given in Fig. S1 in the supplement. The WRF/CUACE v1.0 model uses several modules of the original WRF/Chem model, i.e., modules of advection, vertical mixing, convection, biomass emissions, anthropogenic gas emissions, photolysis and gas dry/wet deposition (Fig. S1). As described in Section 2.2, the CBM-Z mechanism is**
175 **newly added with the KPP protocol (Damian et al., 2002) to replace the RADM2 mechanism in the original CUACE module. An interface procedure, cuace_driver,**

is designed to integrate the core sections of the aerosol physical and chemical processes of the CUACE module with the WRF framework (Fig. S1)."

We have carefully revised the Section 4 for readers to more easily understand how the WRF-Chem model works. For example, we added more description of WRF-Chem: "WRF-Chem is a meteorology-chemistry coupled model. In the chemical module of the WRF-Chem, the processes are split to emissions, vertical mixing, dry deposition, convection, gas chemistry, cloud chemistry, aerosol chemistry and wet deposition, all of which are integrated in an interface procedure (chem_driver). Advection process is treated in the WRF model. Information, such as rainfall rates, vertical mixing coefficients and convective updraft properties, is provided by WRF to calculate the processes treated in the chemical module. WRF-Chem uses registry tools for automatic generation of application code. Physical and chemical variables, as well as options of parameterization schemes are coded in files (such as registry.chem) in the directory of WRFV3/Registry, which provides the convenience for developers to add variables and options.".

[Figure]

**Figure 1.** Schematic of modules in the WRF/CUACE v1.0 system.

*In section 3.2, the authors describe the added heterogeneous chemistry added to the model. I wonder if "Aerosol" stands for all the aerosols in the model, treated the same way or if only a sample of all aerosols are considered in the reaction. Also, the way the reactions are written may let think that the aerosol used as a reactant disappear, or I guess it only acts as a support for the reaction.*

**Response: Sorry for the unclear descriptions. The "Aerosol" stands for all the aerosols in the model. Aerosol in the reactions only acts as a support. We state in the revised Section 3.2 ""Aerosol" in the reactions stands for all the aerosols in the model", and rewrite the forms of heterogeneous chemical reactions from "… + Aerosol → …" to "… $\xrightarrow{\text{Aerosol}}$ …".**

*The description of the model CUACE is not precise enough, essentially concerning the chemical scheme and the reference Zhou et al, (2012) does not either. You claim that RADM2 has 121 reactions, but there are more in Stockwell et al, (1990). Please add the reference for RADM2 and explain the differences between the original publication and your version of RADM2. In section 4, authors explain they added the possibility to use the chemical scheme CBM-Z using KPP. But they do not precise which chemical scheme is finally used. If it is RADM2, then this section should be in the conclusion as future work. If it is CBM-Z then it should be on section 2.2 about CUACE module and more developed: number of species, number of reactions, number of photochemical reactions, way the photochemical reactions are taken into account (especially above the 100hPa upper limit), etc.*

**Response: Thanks very much for pointing out it. We have confirmed with Zhou and checked the code of CUACE. There are totally 136 chemical reactions and 21 photochemical reactions in the RADM2 scheme in CUACE model. We have corrected the mistake in the revised Section 2.2.**

**Sorry for the unclear description of which chemical scheme is finally used. The**

**RADM2 mechanism in the original CUACE module is discarded in the WRF/CUACE v1.0 model, so the CBM-Z mechanism is finally used for simulation. We state in the Section 2.2: "As the gaseous chemistry (RADM2) in the CUACE module is not computationally economic and it is hard coded, which means that it is not conducive to adapting chemical reactions in the future, the CBM-Z photochemical mechanism (Zaveri and Peters, 1999) with a better computational efficiency is added with the KPP protocol (Damian et al., 2002) to replace the RADM2 mechanism.".**

**Following your suggestion, we have revised the section 2.2 to detail the chemical scheme CBM-Z: "CBM-Z mechanism (Zaveri and Peters, 1999) contains 55 species, 114 reactions and 20 photochemical reactions. It is based on the widely used Carbon Bond Mechanism (CBM-IV) and uses the lumped structure approach for condensing organic species and reactions. CBM-Z extends the CBM-IV to include revised inorganic chemistry, explicit treatment of the lesser reactive paraffins, methane and ethane, revised treatments of reactive paraffin, olefin, and aromatic reactions, inclusion of alkyl and acyl peroxy radical interactions and their reactions with $NO_3$, inclusion of organic nitrates and hydroperoxides, and revised isoprene chemistry. Currently, stratospheric chemistry is not included in the CUACE module. Species (i.e, $CH_4$, CO, $O_3$, NO, $NO_2$, $HNO_3$, $N_2O_5$ and $N_2O$) above a specified pressure level are fixed to climatological values. Between the specified pressure level and the tropopause level, the species was relaxed with a 10-day relaxation factor."**

*The present paper deals with a new combination of a NWP and a chemistry model. But only a part of the chemistry is evaluated. It would have been interesting to evaluate the meteorological fields during the simulation made. Moreover the fact that the SCB region seems badly represented for $PM_{2.5}$ is due to the complex terrain could be illustrated.*

**Response: Sincere thanks for the suggestions. The simulated temperature at 2 m**

(T2), relative humidity at 2 m (RH2) and wind speed at 10 m (WS10) were selected for evaluation. Table S1 shows the observation mean, simulation mean, correlation coefficient ($R$), MB, ME and RMSE of the meteorological fields in the NCP, YRD, PRD and SCB, respectively.

The MB and RMSE for T2 vary from 0.48 to 1.14 °C and from 2.01 to 2.50 °C, respectively, indicating surface temperatures are slightly overestimated in the four regions. The $R$ value for T2, ranging from 0.88 to 0.93, indicates the variation trends are well captured by the model. The model underestimates RH2 in the four regions with the MB ranging from -6.22 to -14.30 % and the RMSE ranging from 13.95 to 18.77 %, which are comparable with previous studies in China (Wang et al., 2014; Gao et al., 2016). The RMSE for WS10 in the four regions vary from 1.47 to 1.61 m s$^{-1}$, fall within the "good" model performance criteria (little than 2 m s$^{-1}$) proposed by Emery et al. (2001). However, it should be noted that the $R$ for WS10 in the SCB is relatively poor, indicating the variation trends were not well captured. Generally, the model performed best in the YRD, followed by the PRD and NCP, and performed worst in the SCB for meteorological fields. We have added the evaluation in Section 5.2.1 in the revised manuscript.

**Table S1** Statistical metrics for hourly temperature at 2 m (T2), hourly relative humidity at 2 m (RH2) and hourly wind speed at 10 m (WS10), respectively in the NCP, YRD, PRD and SCB regions.

| | | Obs | Sim | $R$ | MB | ME | RMSE |
|---|---|---|---|---|---|---|---|
| **NCP** | T2 (°C) | 17.31 | 18.07 | 0.91 | 0.76 | 1.87 | 2.34 |
| | RH2 (%) | 62.88 | 51.10 | 0.80 | -11.78 | 14.47 | 17.91 |
| | WS10 (m s$^{-1}$) | 2.05 | 2.99 | 0.64 | 0.95 | 1.29 | 1.60 |
| **YRD** | T2 (°C) | 17.29 | 17.77 | 0.93 | 0.48 | 1.62 | 2.01 |
| | RH2 (%) | 70.74 | 64.51 | 0.82 | -6.22 | 11.28 | 13.95 |
| | WS10 (m s$^{-1}$) | 2.42 | 3.29 | 0.74 | 0.87 | 1.20 | 1.47 |
| **PRD** | T2 (°C) | 22.92 | 24.06 | 0.91 | 1.14 | 2.06 | 2.39 |
| | RH2 (%) | 75.74 | 67.20 | 0.78 | -8.54 | 12.73 | 14.88 |
| | WS10 (m s$^{-1}$) | 2.23 | 3.23 | 0.60 | 1.01 | 1.32 | 1.61 |
| **SCB** | T2 (°C) | 18.02 | 19.53 | 0.88 | 1.52 | 2.04 | 2.50 |
| | RH2 (%) | 74.17 | 59.87 | 0.73 | -14.30 | 15.98 | 18.77 |
| | WS10 (m s$^{-1}$) | 1.35 | 2.05 | 0.40 | 0.70 | 0.99 | 1.24 |

* All $R$ (correlation coefficient) values passed $p < 0.001$.

* Obs and Sim represent the average observations and simulations, respectively.

*In section 5.2, the authors talk about the negative bias in winter in NCP region by saying that the model misses secondary aerosols. But in summer it seems to be a positive bias almost as dramatic as the negative bias in winter. Do the authors have an explanation for this bias?*

**Response: Thanks for pointing out it. According to our analysis, the positive bias in summer in NCP is mainly due to the uncertainty in anthropogenic emissions. It is known that PM$_{2.5}$ concentration is mainly driven by primary emissions, meteorology and chemical reactions. Table S2 shows the statistical metrics for hourly meteorological fields in winter and summer in the NCP. It can be seen that bias of summer meteorological fields is reasonable, and is comparable to those in winter (Table S2) as well as to those in the YRD and PRD (Table S1), which indicate bias in meteorological fields is not the reason. Additionally, In the YRD and PRD, where the uncertainties of anthropogenic emissions are generally known as less than that of NCP, the bias of PM$_{2.5}$ between winter and summer are comparable (Table 3), implying chemical formation of PM$_{2.5}$ in summer is not overestimated by the WRF/CUACE v1.0 model. Therefore, it could be inferred that uncertainties in**

**the emission inventory in the NCP lead to the dramatic positive bias of PM$_{2.5}$. We have added the analysis in Section 5.2.2 in the revised manuscript.**

**Table S2** Statistical metrics for hourly temperature at 2 m (T2), hourly relative humidity at 2 m (RH2) and hourly wind speed at 10 m (WS10), respectively in winter and summer in the NCP.

| NCP region | | Obs | Sim | *R* | MB | ME | RMSE |
|---|---|---|---|---|---|---|---|
| **Winter** | T2 (℃) | 1.59 | 2.01 | 0.85 | 0.42 | 1.67 | 2.14 |
| | RH2 (%) | 62.65 | 53.17 | 0.75 | -9.48 | 14.18 | 18.18 |
| | WS10 (m s$^{-1}$) | 1.82 | 2.64 | 0.62 | 0.82 | 1.15 | 1.46 |
| **Summer** | T2 (℃) | 27.48 | 28.88 | 0.89 | 1.40 | 1.89 | 2.38 |
| | RH2 (%) | 72.79 | 59.61 | 0.84 | -13.18 | 13.98 | 16.42 |
| | WS10 (m s-1) | 1.93 | 2.42 | 0.54 | 0.49 | 1.00 | 1.27 |

**Table 3** Statistical metrics for hourly PM$_{2.5}$ in four haze contaminated areas (2013–2017), in which bold, normal , and italic font for MFB and MFE correspond to the "excellent", "good", and "average" levels in Morris et al. (2005), respectively.

| | R | MB | ME | NMB | NME | MFB | MFE |
|---|---|---|---|---|---|---|---|
| | | μg m$^{-3}$ | μg m$^{-3}$ | % | % | % | % |
| **NCP** | 0.59 | -5.0 | 44.5 | -5.4 | 47.5 | 3.3 | 49.1 |
| Winter | 0.59 | -45.0 | 67.7 | -28.4 | 42.7 | -22.5 | 47.0 |
| Spring | 0.57 | -9.5 | 28.0 | -14.0 | 41.1 | -20.7 | 47.4 |
| Summer | 0.47 | 33.9 | 42.9 | 55.1 | 69.8 | *44.9* | *56.3* |
| Autumn | 0.63 | -0.8 | 39.2 | -0.9 | 45.4 | 9.0 | 45.9 |
| **YRD** | 0.71 | 12.9 | 26.9 | 21.8 | 45.3 | 21.1 | 42.9 |
| Winter | 0.75 | 6.0 | 30.6 | 6.4 | 32.5 | **8.5** | **34.1** |
| Spring | 0.49 | 14.2 | 26.3 | 25.4 | 47.1 | 19.1 | 40.0 |
| Summer | 0.56 | 16.4 | 23.3 | 47.8 | 67.9 | 26.7 | 49.4 |
| Autumn | 0.66 | 15.1 | 27.3 | 28.7 | 51.8 | 29.5 | 48.0 |
| **PRD** | 0.68 | 5.3 | 17.1 | 13.1 | 42.1 | 8.6 | 40.1 |
| Winter | 0.56 | 3.0 | 20.5 | 5.0 | 34.6 | **5.5** | **34.4** |
| Spring | 0.64 | 6.9 | 17.6 | 19.5 | 49.7 | 4.2 | 45.6 |
| Summer | 0.68 | 2.8 | 8.5 | 14.8 | 44.4 | 5.9 | 39.0 |
| Autumn | 0.54 | 8.6 | 21.8 | 17.7 | 45.2 | 18.3 | 41.9 |
| **SCB** | 0.59 | 7.6 | 31.3 | 12.2 | 50.3 | *20.7* | *51.4* |
| Winter | 0.41 | -13.3 | 46.7 | -11.5 | 40.4 | -8.3 | 45.2 |
| Spring | 0.49 | 4.1 | 22.4 | 8.4 | 45.9 | 11.4 | 46.1 |
| Summer | 0.40 | 21.6 | 28.2 | 60.4 | 78.6 | *38.7* | *58.9* |
| Autumn | 0.58 | 15.9 | 28.2 | 31.4 | 55.7 | *37.2* | *54.3* |

*The authors detailed the implementation of the new dry deposition scheme. Also in the conclusion, they wrote "it is difficult to evaluate the dry deposition process is improved", but they did not present any comparison between the two parametrization. A comparison over the already used observed concentrations for the evaluation might be a start for evaluating the improvement.*

**Response: Sincere thanks for the suggestions. We have added the comparison between the two parametrization in the revised Section 3.1 "The most significant difference between the Z01 and PZ10 scheme is the treatment of *Rs*, which stands for the dry velocity contributed by surface resistance, including the effect of Brownian diffusion, turbulent impaction, interception and rebound. According to the study of Wu et al., (2018), dry deposition velocity of fine particles is strongly affected by the Brownian diffusion and turbulent impaction. Thereby, it could be inferred that the Z01 scheme is prone to overestimate the effect of Brownian diffusion and turbulent impaction. In a recent study by Emerson et al. (2020), with observationally constrained approach, the Z01 scheme was revised to be with weaker effect of Brownian diffusion, and as a result, got better performance in simulating the dry deposition velocity of fine particles.".**

**Following the suggestions, we performed simulations for a winter month (January 2015) to show the model improvements with and without the updated dry deposition scheme. As shown in Fig. S3, the $PM_{2.5}$ concentrations were commonly underestimated with the Z01 scheme (Fig. S3a), as it tends to overestimate the dry deposition velocity of fine particles (Petroff and Zhang, 2010). The underestimation was improved significantly when the Z01 scheme was updated to the PZ10 scheme (Fig. S3b). We have added the comparison in the supplementary, and state in the conclusions: "With the observed $PM_{2.5}$ concentrations, model improvements with and without the updated dry deposition scheme are preliminary evaluated (as shown in Figure S3 in the supplement).".**

[Figure]

**Figure S3.** Observed and simulated PM$_{2.5}$ concentrations with (a) Z01 and (b) PZ10 particle dry deposition schemes.

[Figure]

**Figure S2.** The MEIC emissions of $PM_{2.5}$ (a,e,i), $NO_x$ (b,f,j), $SO_2$ (c,g,k), and CO (d,h,l) in the three years of (a-d) 2012, (e-h) 2014, and (i-l) 2016. Emissions outside mainland China is from the MIX emission inventory. The unit is $\mu g\ m^{-2}\ s^{-1}$ for $PM_{2.5}$, $NO_x$, $SO_2$, and is $mg\ m^{-2}\ s^{-1}$ for CO.

*Page 9, line 270: Why do the authors use anthropogenic emissions representative for 2012, 2014 and 2016 to represent the years 2013, 2015 and 2017? Moreover for which year(s) is the MIX inventory representative?*

**Response: Currently, only the MEIC inventory representative for 2012, 2014 and 2016 is open access for download. We use anthropogenic emissions representative for 2012, 2014 and 2016 to represent the years 2013, 2015 and 2017 in order to reflect the changes in anthropogenic emissions. The year of 2010 is the MIX inventory representative for. We have explained this in the revised Section 5.1.**

*Page 10, line 296: Please add the mention 'not shown' for the time series comparison.*

**Response: It has been added.**

*Page 10, line 303: Please add a reference for the aerosol composition.*

**Response: Following the suggestion, the reference (Huang et al., 2014) has been added.**

*Page 11, line 351: Why do the authors only evaluate the simulations against $O_3$ and $NO_2$*
455   *observations? Indeed $SO_2$ observations might be a good observation since it is the direct*
*precursor for sulfate aerosols.*

**Response: Thanks very much for pointing out it. We have added the evaluation of**
**$SO_2$ in Section 5.2.1 in the revised manuscript following the suggestion.**

*Figure 3: (a), (b), (c) and (d) are missing on the figure. The 3 of mg m-3 is not in exponent size.*

460

**Response: All are revised.**

*Table 1: What are the value of γlow and γhigh? What is the value of RHmax? There seems to be a problem at the end of the line with a lonely bracket for the uptake coefficient for $N_xO_y$ and $SO_2$.*

465 **Response: The $\gamma_{low}$ and $\gamma_{high}$ are the lower and upper limits of $\gamma$ values. The *$RH_{max}$* is the *RH* value at which the $\gamma$ reaches the upper limit. The values of $\gamma_{low}$, $\gamma_{high}$ and *$RH_{max}$* are referred to the work of Zheng et al. (2015). That is, values of $\gamma_{low}$ for $N_2O_5$, $NO_2$, $NO_3$ and $SO_2$ are 1E-3, 4.4E-5, 0.1 and 2E-5, respectively corresponding to the values of $\gamma_{high}$ at 0.1, 2E-4, 0.23, 5E-5. The *$RH_{max}$* is 70 % for $N_xO_y$, and is**
470 **100 % for $SO_2$. Thanks for pointing this out. We have added the description $\gamma_{low}$, $\gamma_{high}$ and *$RH_{max}$* in the revised Table.**

*Table 3: Please add "hourly" in the description of the table.*

**Response: It has been added.**

[revised manuscript text omitted]

$$H_2O_2 \text{ (gas)} \xrightarrow{\text{Aerosol}} + \text{Aerosol} \rightarrow \text{Products} \qquad (14)$$

$$HNO_3 \text{ (gas)} \xrightarrow{\text{Aerosol}} + \text{Aerosol} \rightarrow 0.5NO_3^- + 0.5NO_x \text{ (gas)} \qquad (15)$$

$$HO_2 \text{ (gas)} + Fe(II) \rightarrow Fe(III) + H_2O_2 \qquad (16)$$

$$N_2O_5 \text{ (gas)} \xrightarrow{\text{Aerosol}} + \text{Aerosol} \rightarrow 2\,NO_3^- \qquad (17)$$

$$NO_2 \text{ (gas)} \xrightarrow{\text{Aerosol}} + \text{Aerosol} \rightarrow NO_3^- \qquad (18)$$

$$NO_3 \text{ (gas)} \xrightarrow{\text{Aerosol}} + \text{Aerosol} \rightarrow NO_3^- \qquad (19)$$

$$O_3 \text{ (gas)} \xrightarrow{\text{Aerosol}} + \text{Aerosol} \rightarrow \text{Products} \qquad (20)$$

$$OH \text{ (gas)} \xrightarrow{\text{Aerosol}} + \text{Aerosol} \rightarrow \text{Products} \qquad (21)$$

$$SO_2 \text{ (gas)} \xrightarrow{\text{Aerosol}} + \text{Aerosol} \rightarrow SO_4^{2-} \
[revised manuscript text omitted]